



# Methane retrieved from TROPOMI: improvement of the data product and validation of the first two years of measurements

Alba Lorente[1], Tobias Borsdorff[1], Andre Butz[2,3], Otto Hasekamp[1], Joost aan de Brugh[1], Andreas Schneider[1], Frank Hase[4], Rigel Kivi[5], Debra Wunch[6], David F. Pollard[7], Kei Shiomi[8], Nicholas M. Deutscher[9], Voltaire A. Velazco[9], Coleen M. Roehl[10], Paul O. Wennberg[10], Thorsten Warneke[11], and Jochen Landgraf[1]

[1]Earth science group, SRON Netherlands Institute for Space Research, Utrecht, the Netherlands
[2]Institute of Environmental Physics, University of Heidelberg, Heidelberg, Germany
[3]Heidelberg Center for the Environment, University of Heidelberg, Heidelberg, Germany
[4]Institute of Meteorology and Climate Research (IMK-ASF), Karlsruhe Institute of Technology, Karlsruhe, Germany
[5]Greenhouse Gases and Satellite Methods group, Finnish Meteorological Institute, Sodankylä. Finland
[6]Department of Physics, University of Toronto, Toronto, Canada
[7]National Institute of Water and Atmospheric Research Ltd (NIWA), Lauder, New Zealand
[8]Japan Aerospace Exploration Agency (JAXA), Tsukuba, Japan
[11]Institute of Environmental Physics, University of Bremen, Bremen, Germany
[9]Centre for Atmospheric Chemistry, School of Earth, Atmospheric and Life Sciences, University of Wollongong, Wollongong, NSW, 2522
[10]Division of Geological and Planetary Sciences, California Institute of Technology, Pasadena, California, USA

**Correspondence:** Alba Lorente (a.lorente.delgado@sron.nl)

**Abstract.** The TROPOspheric Monitoring Instrument (TROPOMI) aboard of the Sentinel 5 Precursor (S5-P) satellite provides methane ($CH_4$) measurements with high accuracy and exceptional temporal and spatial resolution. TROPOMI $CH_4$ measurements are highly valuable to constrain emissions inventories and for trend analysis, with strict requirements on the data quality. This study describes the improvements that we have implemented to retrieve $CH_4$ from TROPOMI using the RemoTeC

full-physics algorithm. The updated TROPOMI $CH_4$ product features a constant regularization scheme of the inversion that stabilizes the retrieval and yields less scatter in the data, and includes a higher resolution surface altitude database. We have tested the impact of three state-of-the-art molecular spectroscopic databases (HITRAN 2008, HITRAN 2016 and Scientific Exploitation of Operational Missions – Improved Atmospheric Spectroscopy Databases SEOM-IAS) and found that SEOM-IAS provides the best fitting results. The most relevant update in the TROPOMI $XCH_4$ data product is the implementation

of a posteriori correction fully independent of any reference data that is more accurate and corrects for the underestimation at low surface albedo scenes and the overestimation at high surface albedo scenes. After applying the correction, the albedo dependence is removed to a large extent in the TROPOMI versus satellite (Greenhouse gases Observing SATellite – GOSAT) and TROPOMI versus ground-based observations (Total Carbon Column Observing Network – TCCON) comparison, which is an independent verification of the correction scheme. We validate two years of TROPOMI $CH_4$ data that shows the good

agreement of the updated TROPOMI $CH_4$ with TCCON ($-3.4 \pm 5.6$ ppb) and GOSAT ($-10.3 \pm 16.8$ pbb) (mean bias and standard deviation). Low and high albedo scenes as well as snow covered scenes are the most challenging for the $CH_4$ re-



trieval algorithm, and although the posteriori correction accounts for most of the bias, there is a need to further investigate the underlying cause.

## 1 Introduction

Methane ($CH_4$) is the second most important anthropogenic greenhouse gas after carbon dioxide ($CO_2$). The global warming
potential of $CH_4$ for a 20 year horizon is more than 80 times higher than that of $CO_2$ (Myhre et al., 2013), and together with its relatively short lifetime of about 10 years makes it an ideal short-term target for climate change mitigation strategies via reducing $CH_4$ emissions. $CH_4$ has both natural (e.g. wetlands) and anthropogenic sources (e.g. agriculture and waste together with fossil fuels), and globally 60 % of the total emissions are attributed to anthropogenic sources (Saunois et al., 2019). Natural sources are the most uncertain components of the $CH_4$ budget because of their magnitude and variability, which at the same
time depend on characteristics that are vulnerable to changes in the state of the Earth's climate. Furthermore, the interpretation of observed $CH_4$ trends is hampered by the uncertainties in the estimates of $CH_4$ emissions (Turner et al., 2019).

     Satellite observations of $CH_4$ are highly valuable to constrain emission inventories and for trend analysis, not only at global scale but also at regional and local scales. $CH_4$ measurements from satellite instruments like GOSAT (Greenhouse gases Observing SATellite) have been used to infer $CH_4$ emissions from natural sources (e.g. tropical wetlands (Lunt et al., 2019))
and anthropogenic sources (e.g. coal mining in China (Miller et al., 2019)), and also to map emissions and trends at global scale (e.g. Maasakkers et al. (2019)). However, the spatial and temporal resolution at which these emissions can be resolved is limited by the capabilities of the instrument, preventing daily estimations or source attribution at fine scales.

     A unique perspective for the long-term monitoring of $CH_4$ is provided by the TROPOMI (TROPOspheric Monitoring Instrument) instrument on board the Sentinel 5 Precursor (S5-P) satellite with its daily global coverage at an unprecedented
resolution of $7 \times 7$ km$^2$ since its launch in October 2017 (upgraded to $5.5 \times 7$ km$^2$ in August 2019). The high resolution together with the high signal-to-noise ratio allows the detection and quantification of $CH_4$ emissions from localized (e.g. Pandey et al. (2019)) to larger scale sources (e.g. Permian basin by Zhang et al. (2020)). Furthermore, assimilating TROPOMI $CH_4$ has shown great potential (e. g. in the Copernicus Atmosphere Monitoring Service (CAMS) ECMWF Integrated Forecasting System (CAMS-IFS) data assimilation system (Barre et al., 2020)). The main challenge of $CH_4$ remote sensing is that high
data quality is required for data assimilation and flux inversion applications. For TROPOMI, strict mission requirements are formulated with a single sounding precision and accuracy both below 1% (Veefkind et al., 2012).

     TROPOMI $CH_4$ data was already proved to be of good quality by comparisons shortly after launch with both GOSAT observations (Hu et al., 2018) and ground-based measurements from the TCCON network (Hasekamp et al., 2019). However, the $CH_4$ data product can be now further improved using real measurements after TROPOMI has been measuring for more than
two years. A detailed analysis of the data provides insight on which aspects of the processing chain regarding the input data or retrieval algorithm can be further improved. The long-term record also allows to explore possibilities of correcting biases independent of any reference data (e.g. ground-based or other satellite measurements).



In this study we present the improvements that we have developed to retrieve $CH_4$ from TROPOMI measurements using the full-physics approach, and we validate the TROPOMI $CH_4$ product with satellite and ground-based measurements. Section 2 describes the data and analysis methods that we use and Sect. 3 focuses on the main improvements related to the regularization scheme of the inversion, the choice of the spectroscopic database for the absorption cross sections, the surface elevation

database and a posteriori bias correction derived using only TROPOMI data. Finally, Sect. 4 and Sect. 5 show a detailed validation of the improved TROPOMI $CH_4$ data. The study concludes in Sect. 6 with an outlook for future steps regarding $CH_4$ data retrieved from TROPOMI.

## 2 Retrieval algorithm and validation data sets

### 2.1 TROPOMI $CH_4$ retrieval algorithm

The methane total column-average dry-air mole fraction ($XCH_4$) is retrieved from TROPOMI measurements of sunlight backscattered by Earth's surface and atmosphere in the near-infrared (NIR) and shortwave-infrared (SWIR) spectral bands with the retrieval algorithm RemoTeC. This algorithm has been extensively used to retrieve both $CO_2$ and $CH_4$ from measurements of OCO-2 and GOSAT (e.g. Wu et al. (2018); Butz et al. (2011)) and it is the Sentinel 5-P and Sentinel 5 operational algorithm for $CH_4$ (Hasekamp et al. (2019); Landgraf et al. (2019)).

The S5P RemoTeC algorithm uses the full-physics approach that simultaneously retrieves the amount of atmospheric $CH_4$ and the physical scattering properties of the atmosphere. The algorithm aims at inferring the state vector $\boldsymbol{x}$ that contains all the parameters to be retrieved from the radiance measurements $\boldsymbol{y}$ in the SWIR (2305-2385 nm) and NIR (757-774 nm) spectral bands, where the forward model $\boldsymbol{F}$ simulates the TROPOMI measurements,

$$\boldsymbol{y} = \boldsymbol{F}(\boldsymbol{x}) + \epsilon_y + \epsilon_F. \tag{1}$$

Here, $\epsilon_y$ and $\epsilon_F$ are the measurement noise error and the forward model error respectively. The forward model employs the LINTRAN V2.0 radiative transfer model in its scalar approximation to simulate atmospheric light scattering and absorption in a plane parallel atmosphere (Schepers et al. (2014); Landgraf et al. (2001)). Accurate modelling of absorption by molecules relies on spectroscopic databases, which provide the absorption cross-section of the target absorber $CH_4$ as well as of the interfering gases CO, $H_2O$ and $O_2$.

The inversion to estimate the state vector $\boldsymbol{x}$ requires the use of regularization methods, as measurements typically do not contain sufficient information to retrieve every state vector element independently. The RemoTeC retrieval algorithm uses the Philips-Tikhonov regularization scheme, which aims to find the state vector by solving the minimization problem

$$\hat{\boldsymbol{x}} = \min\left(||\boldsymbol{S}_y^{-1/2}(\boldsymbol{F}(x) - \boldsymbol{y})||^2 + \gamma||\boldsymbol{W}(\boldsymbol{x} - \boldsymbol{x}_a)||^2\right), \tag{2}$$





where $|| \cdot ||$ describes the Euclidian norm, $\boldsymbol{S}_y$ is the measurement error covariance matrix that contains the noise estimate, $\gamma$ is the regularization parameter, $\boldsymbol{W}$ is a diagonal weighting matrix that renders the side constraint dimensionless and ensures that only the target absorber $CH_4$ and the scattering parameters contribute to its norm (Hu et al., 2016), and $\boldsymbol{x}_a$ is the a priori state vector.

The retrieval state vector contains $CH_4$ partial sub-column number densities at 12 equidistant pressure layers. The total column of the interfering non-target absorbers CO and $H_2O$ are also retrieved, together with the effective aerosol total column, size and height parameter of the aerosol power law distribution. A Lambertian surface albedo in both NIR and SWIR spectral range together with its first order spectral dependence is also retrieved, as well as spectral shift and fluorescence in the NIR band.

The TROPOMI $CH_4$ data product is given in the form of total column-averaged dry-air mole fraction, $XCH_4$. It is calculated from the methane vertical subcolumn elements $x_i$ and the dry air column $V_{\mathrm{air,dry}}$ calculated with meteorology input from ECMWF (European Centre for Medium-Range Weather Forecasts) analysis product and surface topography from a high resolution database:

$$XCH_4 = \sum_{i=0}^{n} \frac{x_i}{V_{\mathrm{air,dry}}}. \tag{3}$$

The precision $\sigma_{XCH_4}$ is given as the standard deviation of the retrieval noise, which follows from the error covariance matrix $\boldsymbol{S}_x$ that describes the effect of the measurement noise on the retrieval (Hu et al., 2016):

$$\sigma_{XCH_4} = \frac{\sqrt{\sum_{i,j=0}^{n} S_{x,i,j}}}{V_{\mathrm{air,dry}}} \tag{4}$$

The algorithm has been designed to provide accurate and precise retrievals for clear-sky scenes with minor scattering by aerosols and optically thin cirrus. To fulfill this criterion, a strict cloud filter is applied based on observations of the Visible
Infrared Imaging Radiometer Suite (VIIRS) aboard the Suomi-NPP satellite that observes the same scene as TROPOMI approximately 5 minutes earlier. In cases when VIIRS data is not available, we use a back-up filter based on a non-scattering $H_2O$ retrieval from the weak and strong absorption bands (Hu et al., 2016). Table A1 summarizes the filters applied in the retrieval process and in the TROPOMI data selection used in this study.

The $CH_4$ total column-average dry-air mole fraction retrieved from TROPOMI with the operational retrieval algorithm
(version 1.2.0 as of June 2020) largely complies with the mission requirement of precision and accuracy below 1%, with significantly improved data quality of the bias-corrected product (Hasekamp et al., 2019). In Sect. 3 we present recent updates that further improve the quality of the data. This updated retrieval algorithm is referred to as the beta-version of the TROPOMI $XCH_4$ data product.



## 2.2 TCCON reference dataset

To validate XCH$_4$ retrieved from TROPOMI we use independent ground-based XCH$_4$ measurements from the Total Carbon Column Observing Network (TCCON) (Wunch et al., 2011a) as a reference (data version GGG2014). Table 1 contains the information of the 13 different stations located in North America, East Asia, Europe and Oceania used for the validation. In

regions where there are multiple TCCON stations, we have selected those located at flat terrain in relatively remote areas, which minimizes the errors due to assumptions on the vertical CH$_4$ distribution used to correct for differences between the surface elevation of TROPOMI particular pixels and the ground altitude at the TCCON sites.

**Table 1.** Overview of the stations from the TCCON network used in this study.

| Site (Country) | Coordinates Lat, Lon (°) | Altitude (m.a.s.l.) | Reference |
|---|---|---|---|
| **Sodankylä** (Finland) | 67.37, 26.63 | 190 | Kivi and Heikkinen (2016) Kivi et al. (2017) |
| **East Trout Lake** (Canada) | 54.36, -104.99 | 500 | Wunch et al. (2017) |
| **Karlsruhe** (Germany) | 49.1, 8.44 | 110 | Hase et al. (2017) |
| **Orléans** (France) | 47.97, 2.11 | 130 | Warneke et al. (2017) |
| **Park Falls** (US) | 45.94, -90.27 | 440 | Wennberg et al. (2017a) |
| **Lamont** (US) | 36.6, -97.49 | 320 | Wennberg et al. (2017b) |
| **Pasadena** (US) | 34.14, -118.13 | 240 | Wennberg et al. (2017c) |
| **Edwards** (US) | 34.95, -117.88 | 30 | Iraci et al. (2016) |
| **Saga** (Japan) | 33.24, 130.29 | 10 | Kawakami et al. (2017) |
| **Darwin** (Australia) | -12.46, 130.93 | 30 | Griffith et al. (2017a) |
| **Wollongong** (Australia) | -34.41, 150.88 | 30 | Griffith et al. (2017b) |
| **Lauder**\* (New Zealand) | -45.04, 169.68 | 370 | Sherlock et al. (2017) Pollard et al. (2019) |

\*For the Lauder station the *ll* instrument was replaced on October 2018 to *ll*.

To evaluate the quality of the retrieved TROPOMI XCH$_4$, we average TROPOMI XCH$_4$ data within a collocation radius around each station of 300 km. The average retrieved TROPOMI XCH$_4$ within the specific radius is compared with daily

average measurements of the matching TCCON station (i.e. no time constraint) (XCH$_{4,\text{TROPOMI}}$ − XCH$_{4,\text{TCCON}}$). For all paired collocations at each station, we compute the mean bias defined as the mean of the difference of individual collocations ($\Delta$CH$_4$) and its standard deviation ($\sigma$) as a measure of the spread in the data. We then compute the average of the station biases ($\bar{b}$) and its standard deviation ($\sigma(\bar{b})$) as a measure of the station-to-station variability. The station-to-station variability is an important diagnostic parameter as it indicates regional biases in our data, and it might be used as an overall uncertainty

estimate.





## 2.3 GOSAT reference dataset

$XCH_4$ measurements by The Thermal And Near infrared Sensor for carbon Observation - Fourier Transform Spectrometer (TANSO-FTS) on board the Greenhouse gases Observing SATellite (GOSAT) satellite are used for the validation of the TROPOMI $XCH_4$ data. GOSAT was launched in 2009, and with a swath of 790 km and 10.5 km resolution, global coverage is
obtained every 3 days.

We use the GOSAT proxy $XCH_4$ data product produced at SRON in the context of the ESA GreenHouse Gas Climate Change Initiative (GHG CCI) project (Buchwitz et al., 2019, 2017). This $XCH_4$ product is retrieved using the RemoTeC/proxy retrieval algorithm. The proxy approach (Frankenberg et al., 2005) infers a $CO_2$ and $CH_4$ total column from observations at 1.6 μm ignoring any atmospheric scattering in the retrieval. Substantially, the $XCH_4$ product is derived by

$$XCH_4{}^{proxy} = \frac{V_{CH_4}}{V_{CO_2}} \cdot XCO_2{}^{mod} \tag{5}$$

where the column-average dry-air mole fraction $XCO_2{}^{mod}$ is taken from the Carbon Tracker data assimilation system, and $V_{CH_4}$ and $V_{CO_2}$ are the vertical column densities. This approach assumes that light path modifications due to scattering in the atmosphere are the same for the target absorber (i.e. $CH_4$) and the proxy absorber $CO_2$, whose prior is assumed to be known with high accuracy.

The proxy approach cannot be applied to retrieve $XCH_4$ from TROPOMI since it does not cover the 1.6 μm $CH_4$ and $CO_2$ absorption bands. Schepers et al. (2012) compared both the physics and proxy retrievals applied to GOSAT measurements to retrieve $XCH_4$ and concluded that both retrievals performed similarly when validating the retrieved $XCH_4$ with ground-based TCCON measurements.

## 3   TROPOMI $CH_4$ retrieval updates

Our updated TROPOMI $XCH_4$ product corresponds to the S5P-RemoTeC algorithm version 1.3.0 that will be suggested for use in the operational processing (Hu et al. (2016), data product 1.2.0) in the next processor update. The updates to the S5P-RemoTeC retrieval algorithm relate to the regularization scheme, the selection of the spectroscopic database, the implementation of a higher resolution digital elevation map (DEM) for surface altitude and a more sophisticated a posteriori correction for the albedo dependence. In this section we present the updates and quantify the improvements, and we use the comparison with
TCCON and GOSAT as a benchmark to test the performance of the retrieval after implementing the updates.

## 3.1   Regularization scheme

Hu et al. (2016) determined the regularization parameter $\gamma$ in the inversion (Eq. 2) using the L-curve criterion (Hansen (1998), Hu et al. (2016)) in each iteration of the TROPOMI measurement inversion. As TROPOMI has been measuring for more than two years, it is possible to select a constant regularization optimized for real observations. This includes a dedicated





regularization parameter for the target absorber $CH_4$ and one for each of the aerosol parameters (aerosol distribution height and size parameter, and aerosol column). The advantage of the constant regularization is a more stable performance compared to the L-curve method in which the regularization strength changes at each iteration for every scene. The regularization parameters are selected such that the degrees of freedom for $CH_4$ are between 1 and 1.5 and that retrieved aerosol parameters have realistic

distributions.

The main improvement of the constant regularization is that the dispersion in the retrieved $XCH_4$ is significantly reduced. This is noticeable in the $XCH_4$ distribution over small regions where we do not expect large gradients of $XCH_4$. At regions with relatively low albedo, the decrease in the spread of the data can reach 10-20 % (e.g. from 18 ppb to 14 ppb over Canada and 11 to 9 over Australia). Furthermore, the validation with TCCON shows a decrease in the station-to-station variability

of 4 ppb (25 % decrease of the 15 ppb station-to-station variability using the L-curve approach) when analysing one year of data. The comparison with GOSAT shows that the new regularization scheme reduces the standard deviation of the difference between collocated GOSAT and TROPOMI $XCH_4$ observations by 9 % (19.7 ppb to 24.5 ppb).

## 3.2   Spectroscopy database

The TROPOMI $CH_4$ retrieval uses external spectroscopic information to simulate the molecular absorption lines of the target

absorber $CH_4$ as well as of CO and $H_2O$. The baseline retrieval algorithm employs the HITRAN 2008 spectroscopic database (Rothman et al., 2009) with updated spectroscopy parameters for $H_2O$ from Scheepmaker et al. (2013). In preparation for the Sentinel 5 Precursor mission, Birk et al. (2017) established an improved spectroscopic database, the so-called "Scientific Exploitation of Operational Missions - Improved Atmospheric Spectroscopy Databases" (SEOM-IAS hereafter) for the interpretation of TROPOMI observations. The release of the HITRAN 2016 database already included some of the updates from

the SEOM-IAS project regarding $H_2O$ (Gordon et al., 2017). We have tested the effect of the three spectroscopic databases on the retrieved $XCH_4$ using one year of TROPOMI data (Sep 2018 - Sep 2019).

The TCCON validation shows that after substituting HITRAN 2008 by HITRAN 2016 and SEOM-IAS for all the molecules in the $CH_4$ retrieval, the station-to-station variability does not change significantly (less than 1 ppb, see Table 2). The change in the mean bias shows that the different spectroscopy databases introduce an overall bias in the retrieved $XCH_4$ with respect

to HITRAN 2008 (+15.5 pbb for HITRAN 2016 and $-14.7$ ppb for SEOM-IAS), but the correlation of the bias with other retrieved parameters (surface albedo, $H_2O$) does not improve or worsen with any of the spectroscopic database. The spectral fitting quality parameters (e.g. the root mean square of the spectral fit residuals (RMS) and the corresponding $\chi^2$) show a slight improvement over TCCON stations when using the SEOM-IAS spectroscopic database, similar to what was found for the CO retrieval from TROPOMI (Borsdorff et al., 2019). The comparison with $XCH_4$ measured by GOSAT also shows that different

spectroscopic database introduce an overall bias but the standard deviation of the bias does not change significantly (Table 2).

On a global scale, we see that both the RMS and $\chi^2$ improve significantly when using the SEOM-IAS database, with HITRAN 2008 giving the worst fitting results. Figure 1 shows the latitudinal distribution of $XCH_4$ retrieved with HITRAN 2008, SEOM-IAS and HITRAN 2016, referenced to the value at 0° latitude. $XCH_4$ retrieved with HITRAN 2016 shows the least latitudinal variation at latitudes higher than 55° where differences between the datasets are largest, however the global distri-

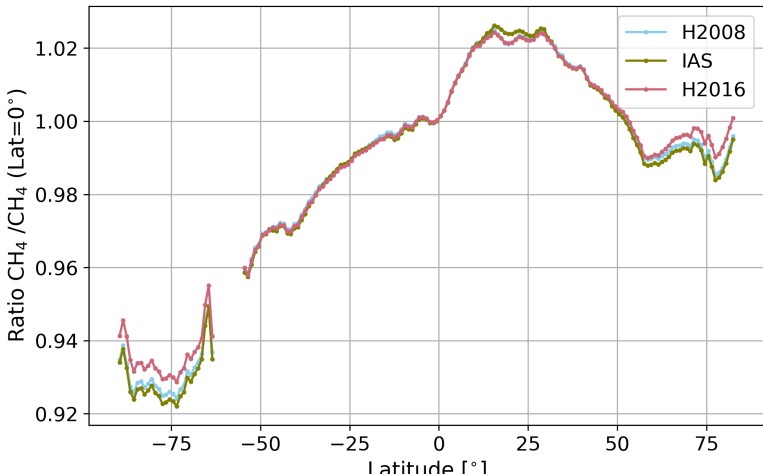

**Figure 1.** Latitudinal distribution of TROPOMI XCH$_4$ retrieved using HITRAN 2008 (blue), HITRAN 2016 (pink) and SEOM-IAS (green), referenced to the value at 0° latitude. Daily measurements from Sep 2018 – Sep 2019 are gridded into a 0.2° x 0.2° grid, averaged longitudinally and then binned in 1° latitude.

bution does not point to a better performance of any of the spectroscopic database. The validation with TCCON observations including Eureka (80.05°N) and Lauder (45.04° S) just reflects the overall bias, but does not point to any latitudinal bias of XCH$_4$ retrieved with any of the spectroscopic database (not shown).

The results of the sensitivity tests do not point to an improved data quality when HITRAN 2016, SEOM-IAS or HITRAN 2008 spectroscopic database is used. Each of them introduces an overall bias that cannot be used as an independent argument to favour a specific database. In view of the better spectral fitting results in the retrieved XCH$_4$ we have decided to use the SEOM-IAS spectroscopy database.

### 3.3 Surface elevation

Satellite remote sensing of XCH$_4$ requires accurate knowledge of surface pressure and thus of surface elevation, which is specially relevant for the spatially highly resolved measurements of TROPOMI. The effect is two-fold: (1) through the pressure dependence of the absorption cross sections and (2) through the dry air column used to calculate dry air mixing ratio from the retrieved column (Eq. 3).

In a first pre-processing step of the retrieval, the elevation data from a digital elevation map (DEM) is interpolated in space to the ground pixel. Then a correction is applied to the atmospheric variables (i.e. surface pressure and model pressure levels) based on the difference between the coarse resolution ECMWF altitude and the surface elevation from the DEM. To minimize errors, a filter is applied on terrain roughness, which excludes scenes with a standard deviation of the surface elevation higher than 80 m within the observed area. The default source for surface elevation information for all TROPOMI products is the



**Table 2.** Overview of the TCCON and GOSAT validation results (mean bias and its standard deviation) for the TROPOMI XCH$_4$ retrieved with different spectroscopic databases.

|  | $\bar{b} \pm \sigma(\bar{b})$ CH$_4$[ppb]* |
| --- | --- |
| **TCCON** |  |
| HITRAN 2008 | $-2.4 \pm 11.7$ |
| SEOM-IAS | $-17.1 \pm 12.4$ |
| HITRAN 2016 | $17.9 \pm 11.1$ |
| **GOSAT** |  |
| HITRAN 2008 | $3.9 \pm 20.1$ |
| SEOM-IAS | $-8.4 \pm 22.8$ |
| HITRAN 2016 | $23.8 \pm 19.7$ |

\* $b =$ TROPOMI $-$ref

Global multi-resolution terrain elevation data 2010 DEM (GMTED2010) with an aggregation radius of 5 km and a sampling of around 2 km, which results in a resolution of approximately 2 km (S5P-DEM hereafter).

The updated retrieval scheme uses the Shuttle Radar Topography Mission (SRTM) (Farr et al., 2007) digital elevation map with a resolution of 15 arcsec, approximately 400 meters. To match the DEM surface elevation with the ground pixel, we
perform a spatial sampling of 0.5 km and compute the mean altitude and its standard deviation for each scene. Figure 2 (upper panel) shows altitude differences between S5P-DEM and SRTM collocated to TROPOMI pixels (before altitude correction) on 5 May 2019 over the United States. In this specific area, 5 % of the pixels have differences in altitude greater than 45 m, with the highest differences over mountain regions. For these scenes the differences in retrieved XCH$_4$ are up to 7 pbb. On a yearly average (and after correction and quality filtering), 1 % of the retrievals present altitude differences greater than 50
m, which result in surface pressure differences above 5 hPa and XCH$_4$ differences above 10 ppb (Fig. 2 lower panels). The terrain roughness within TROPOMI pixels has not changed significantly with the SRTM DEM, so it does not affect the data yield due to the 80 m threshold. Although globally the average altitude difference is small, the analysis of small scale XCH$_4$ enhancements will benefit from this update. Due to its higher resolution the SRTM DEM is a better representation of elevation not only over mountains, but also close to coastlines and over rough terrain (e.g. Greenland, Sahara desert).

**3.4 Posteriori correction**

Greenhouse gas concentrations retrieved from satellite instruments like TROPOMI generally show systematic biases with different instrumental or geophysical parameters. Retrieved CO$_2$ and CH$_4$ from GOSAT and OCO-2 are typically corrected for dependencies on goodness of fit, surface albedo or aerosol parameters (e.g. Guerlet et al. (2013), Inoue et al. (2016), Wu et al. (2018)). In the approach that O'Dell et al. (2018) derived for OCO-2 CO$_2$ retrievals, such parametric bias is part of a more
complex correction that also accounts for footprint-level and global biases using a set of four "truth proxies" as a reference.

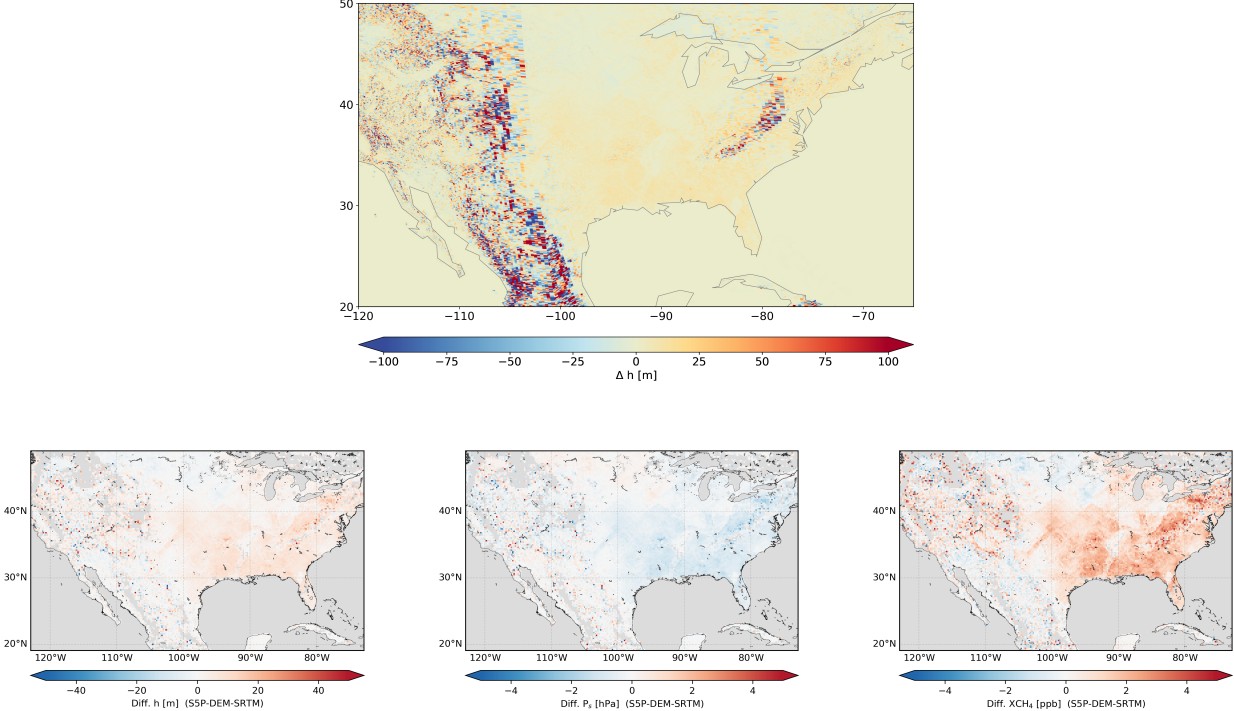

**Figure 2.** Upper panel: altitude difference between S5P-DEM and SRTM collocated to TROPOMI pixels on 5 May 2019 (orbits 8077, 8078, 8079). Lower panel: altitude, surface pressure and XCH$_4$ differences averaged over a year, with custom quality filtering for the TROPOMI XCH$_4$ retrievals, in a 0.2° x 0.2° grid over United States (20-50N, 65-120W).

The comparison of TROPOMI and TCCON XCH$_4$ measurements shows a dependence of the bias (i.e. difference between TROPOMI and TCCON) on surface albedo, while for the other retrieved parameters the dependence is negligible (compared to that of the surface albedo, see Fig. 3). Figure 3a shows that for low albedo values, TROPOMI XCH$_4$ strongly underestimates TCCON measurements, while for relatively high albedo values TROPOMI overestimates TCCON measurements. The comparison of TROPOMI XCH$_4$ with XCH$_4$ retrieved from measurements of GOSAT shows the same dependence of the bias with the retrieved surface albedo. For scenes with low albedo values, generally the retrieval's sensitivity is low due to the large measurement noise. therefore errors from unaccounted light path modification due to scattering processes can be more significant than for scenes with a relatively higher albedo. For low albedo scenes, this effect leads to an underestimation in the retrieved trace gas (Guerlet et al. (2013); Aben et al. (2007)), resembling the TROPOMI XCH$_4$ underestimation in Fig. 3a.

To account for the albedo dependence, we apply an a posteriori bias correction to the retrieved XCH$_4$. In the baseline algorithm, we applied a correction based on the comparison of TROPOMI XCH$_4$ with GOSAT retrievals (Hasekamp et al., 2019). After more than two years of measurements, we have sufficient data to derive the correction using only TROPOMI XCH$_4$ measurements. We use a similar approach to the "small area approximation" applied to OCO-2 (O'Dell et al., 2018), assuming





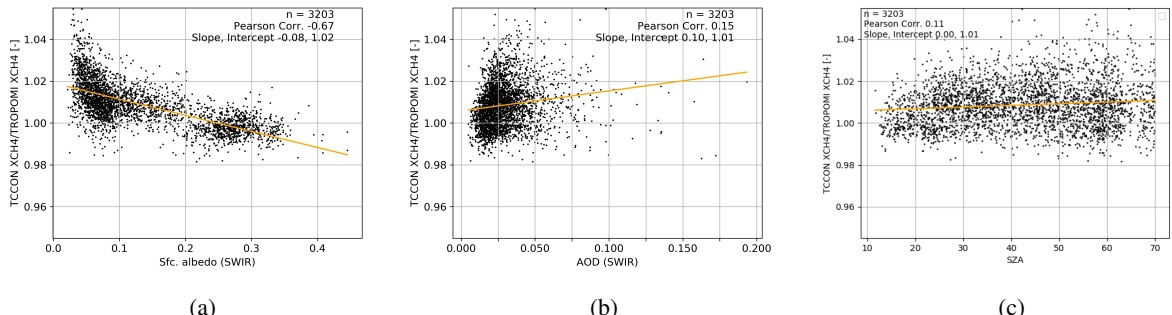

**Figure 3.** Ratio of XCH$_4$ measurements by TCCON and TROPOMI as a function of (a) retrieved surface albedo in the SWIR spectral range, (b) retrieved effective aerosol optical depth (AOD) in SWIR spectral range and (c) solar zenith angle (SZA).

a uniform XCH$_4$ distribution as a function of albedo in several regions. This approach makes the correction completely independent of any reference data (e.g. GOSAT, TCCON) that could introduce additional biases when applying the correction and does not allow for an independent verification of the correction.

The new correction is derived as follows:

1. We select areas at several latitudes and longitudes throughout the globe, small enough so we can assume that XCH$_4$ does not vary, but large enough to cover scenes with a wide range of albedo values. Figure B1 shows the different regions.

   2. For each region we estimate a XCH$_4$ reference value for a surface albedo around 0.2 and then we calculate the ratio of the retrieved XCH$_4$ to the reference value to obtain the albedo dependence. The specific value for surface albedo is selected because XCH$_4$ retrieval errors are lower in the SWIR for that albedo range: errors because of unaccounted light

path modifications due to scattering and surface albedo are minimal around a surface albedo of 0.2 (e.g. Guerlet et al. (2013); Aben et al. (2007)).

   3. We combine the albedo dependence for all the areas, we fit the curve using B-spline interpolation and least squares fitting.

The B-spline method fits piece-wise polynomials that are continuous at the pre-selected knots. The knots and the order of

the polynomials are chosen such that the residual RMS of fit residuals is minimum and that the shape of the fit at the edges of the surface albedo range does not vary sharply.

Figure 4 shows the distribution of the reference to TROPOMI XCH$_4$ ratio for all the areas and the result of the B-spline fit. We observe two distinct features: (1) the strong underestimation for low albedo values (already shown in the TCCON comparison in Fig. 3a), for which the B-spline fit corrects more strongly than the regular polynomial fit that was previously

used and (2) an overestimation for high albedo values, not captured by TCCON due to the limited albedo range values but reported in the TROPOMI and GOSAT comparisons.





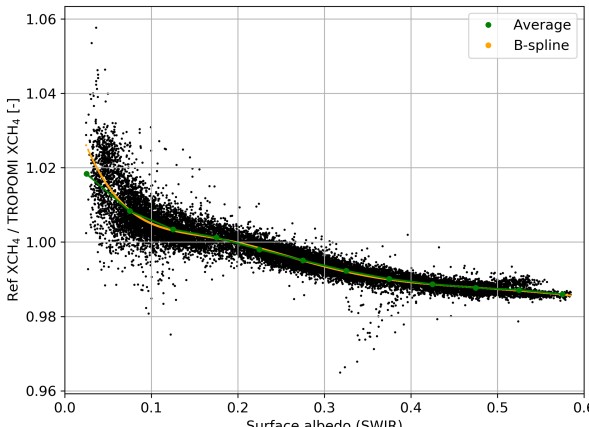

**Figure 4.** Ratio of reference XCH$_4$ to TROPOMI XCH$_4$ as a function of the retrieved surface albedo as explained in step 3 in the derivation of the bias correction. Green dots show the average ratio in 0.05 albedo bins and orange line shows the B-spline fit used to derive the bias correction. Data is averaged from 1 Jan 2018 until 31 Dec 2019 in a $0.1°$ x $0.1°$ grid.

The correction applied to the retrieved XCH$_4$ can be expressed as:

$$\mathrm{XCH}_{4\,i}^{\mathrm{corr}} = \mathrm{XCH}_{4\,i} \cdot f\left(A_{\mathrm{s}\,i}\right). \tag{6}$$

The correction function $f$ depends on the retrieved surface albedo $A_{\mathrm{s}}$ at each pixel $i$.

Figure 5 shows the global distribution of XCH$_4$ before and after applying the correction. Distinctive features that correspond
with low and high surface albedo areas are visible in the difference map. After correction, for example, the XCH$_4$ underestimation for low albedo values (e.g. over high latitudes over Canada and Russia) is corrected. Similarly, the XCH$_4$ overestimation for high albedo values over desert areas like Sahara is accounted for in the correction. The change in XCH$_4$ induced by the bias correction is in the range of 2 %, in agreement with the errors observed in the TCCON comparison.

As the correction is derived using only TROPOMI XCH$_4$ data, the comparison with TCCON and GOSAT is an independent
verification of the approach. The validation with TCCON shows a reduction of 5.9 ppb (50%) in the station-to-station variability and of 13.6 ppb in the bias due to the albedo correction. The dependence of the bias on surface albedo is removed (Fig. 3a vs. Fig. 6a) and the dependence on other parameters remains negligible (not shown). The comparison with GOSAT measurements shows that bias dependence on albedo is removed after applying the correction (Fig. 6b), which reduces by 4 ppb the scatter of the differences in XCH$_4$ measured by the two satellites. In the remainder of the paper the corrected XCH$_4$ product will be
used.





(a)

(b)

(c)

**Figure 5.** (a) Global TROPOMI $XCH_4$ distribution before correction, (b) after correction and (c) their difference ($XCH_4$-$XCH_4^{corr}$) for 2019 averaged in a cylindrical equal-area grid with $0.3°$ x $0.5°$ resolution at the Equator.

## 4 Comparison of TROPOMI and TCCON

### 4.1 TCCON validation

We perform a detailed comparison of the TROPOMI $XCH_4$ corrected with $XCH_4$ measured at 13 TCCON stations selected for the validation (Table 1). TROPOMI is able to capture the temporal $XCH_4$ variability, both the seasonal cycle and the year-

5 to-year increase. This is clearly visible in the time series (e.g. Pasadena or Lamont) in Fig. 7, which shows the time series of





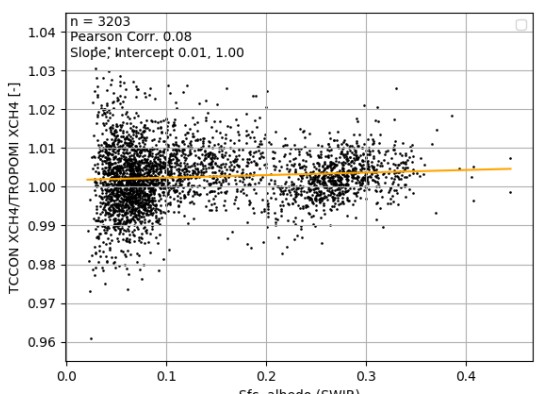
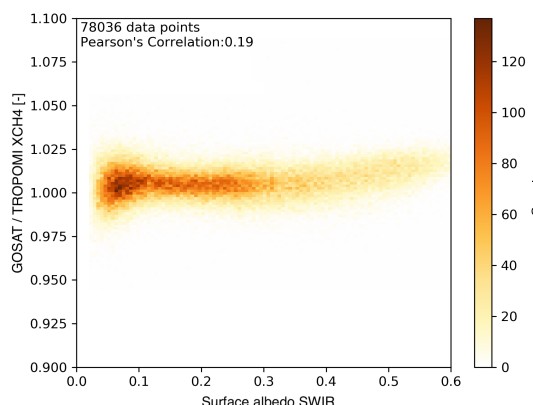

**Figure 6.** Ratio of daily XCH$_4$ measurements by (a) TCCON and TROPOMI and (b) GOSAT and TROPOMI as a function of retrieved surface albedo in the SWIR spectral range. Data for the period 1 Dec 2018 - 31 Dec 2019 is shown.

daily average XCH$_4$ measured at each TCCON station and by TROPOMI for the period 1 Dec 2018 – 31 Dec 2019, with a collocation radius of 300 km.

The mean bias is below 1 % for all stations; the validation results are summarized in Table 3. The average bias for all stations is -0.2 % ($-3.4$ pbb) and the station to station variability is 0.3% (5.6 pbb), both parameters below the mission requirements

for TROPOMI XCH$_4$ retrievals. Compared to the uncorrected TROPOMI XCH$_4$, the mean bias is reduced significantly (from $-3.4$ % to 0.2 %) even though the correction approach does not include any term to correct a global bias. As the overall negative bias is driven by the strong XCH$_4$ underestimation for low albedo values (Fig. 3a), correcting for the albedo bias partly accounts for the overall bias.

Figure 8a shows the mean bias and the standard deviation for each of the stations and Fig. 8b shows the correlation plot. For

a more strict collocation criterion of 100 km radius instead of 300 km, the number of points is reduced significantly but the results of the validation do not change.

## 4.2 High latitude stations

Measurements at high latitude stations such as East Trout Lake (54.36°N) and Sodankylä (67.37°N) show the highest variability and the highest bias in the validation before correction, which is partially reduced by the albedo correction (see validation results

in Table 3). There is a seasonality in the bias which is positive during February – April period and changes to a negative bias around May that then increases to reach the highest (negative) values in fall. This seasonality can be attributed to the fact that during the winter there is snow in these regions at high latitudes as a result of cold, dry air, influencing XCH$_4$ measurements by TROPOMI that affect the validation with TCCON measurements.

Figure 9 shows the time series of the bias between TROPOMI and TCCON XCH$_4$ together with the surface albedo retrieved

in both the SWIR and NIR spectral range over East Trout Lake, Sodankylä and Lamont, the latter included as a mid-latitude





**Figure 7.** Time series of daily averaged XCH$_4$ measurements from TROPOMI (red) and TCCON (blue) over the selected stations for the period 1 Dec 2018 – 31 Dec 2019. TROPOMI measurements around a circle of 300 km radius around each station have been selected for the comparison.





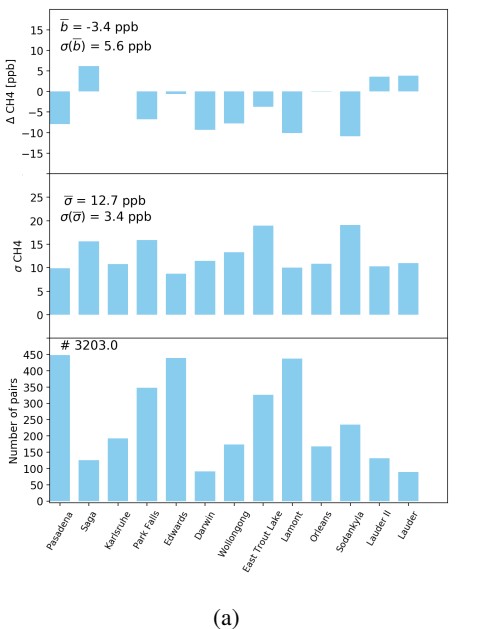
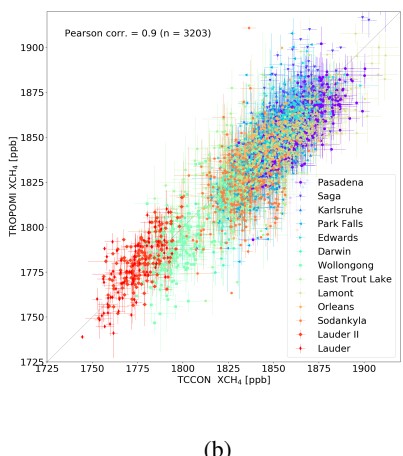

(a)                                                 (b)

**Figure 8.** (a) Mean differences between TROPOMI and TCCON $XCH_4$ ($\Delta XCH_4$), the standard deviation of the differences ($\sigma_{XCH_4}$) and the number of collocations for each of the stations selected for the validation. (b) Correlation of daily average $XCH_4$ measured by TROPOMI and TCCON for all the stations.

reference. Low surface albedo in the SWIR together with high surface albedo in the NIR indicates the presence of snow which is highly correlated with the seasonality in the bias in East Trout Lake and Sodankylä, seasonality that is more pronounced in 2019 than 2018. The seasonal bias is also correlated with high hydrogen fluoride (HF) and low $H_2O$ concentrations (not shown). High HF concentrations are an indication of the influence of the vortex in a specific location, as HF is mostly found in

the stratosphere; HF together with the contrast between surface albedo retrieved in the SWIR and NIR spectral ranges can be used as a proxy to identify the presence of snow and dry air from dynamic meteorological situations at high latitudes.

The presence of snow at high latitude stations shifts the focus to retrieval errors as the most probable cause of the seasonal bias between TCCON and TROPOMI, rather than errors due to collocation or influence of the different priors. Scenes covered by snow are characterized by low spectrum intensity in the SWIR, so signal-to-noise ratio is a limiting factor. On the other

hand, the high TROPOMI signal in the NIR suggests that the weighting of each band might not be optimal in the inversion. As the optical properties are different in the NIR and SWIR bands, errors in the quantification of light path modifications over snow covered scenes can lead to an overestimation of retrieved $XCH_4$. Furthermore, if $H_2O$ may compensate for any radiometric offset in the strong $CH_4$ absorption bands, then in such dry conditions this would not be as effective in winter as in spring–fall, causing the seasonality on the bias. A high bias in high latitudes correlated with $H_2O$ columns was also found

in $H_2O$/HDO retrievals from TROPOMI by Schneider et al. (2020). Note that the seasonal bias is also present when $XCH_4$ is retrieved using the spectroscopic databases discussed in Sect. 3.2.





**Table 3.** Overview of the validation results of TROPOMI XCH$_4$ with measurements from the TCCON network at selected stations. The table shows number of collocations, mean bias and standard deviation for each station and the mean bias for all stations and the station-to-station variability. Results are shown for TROPOMI XCH$_4$ with and without the albedo bias correction applied.

| Site, Country, Lat-Lon Coord. | Nr. of points | Corrected TROPOMI XCH$_4$ and TCCON | | Uncorrected TROPOMI XCH$_4$ and TCCON | |
| --- | --- | --- | --- | --- | --- |
| | | Bias [ppb] (%) | Standard deviation [ppb] (%) | Bias [ppb] (%) | Standard deviation [ppb] (%) |
| **Pasadena** (US) (34.14, −118.13) | 399 | −8.0 (−0.4) | 9.8 (0.5) | 0.6 (0.03) | 9.3 (0.5) |
| **Saga** (Japan) (33.24, 130.29) | 117 | 6.2 (0.3) | 15.6 (0.8) | −17.6 (−0.9) | 13.0 (0.7) |
| **Karlsruhe** (Germany) (49.1, 8.44) | 196 | 0.02 (0.0) | 10.8 (0.6) | -19.2 (−1.0) | 10.1 (0.5) |
| **Darwin** (Australia) (−12.46, 130.93) | 93 | −9.3 (−0.5) | 11.4 (0.6) | −16.5 (−0.9) | 11.8 (0.7) |
| **Wollongong** (Australia) (−34.41, 150.88) | 132 | −7.8 (−0.4) | 13.3 (0.7) | −19.6 (−1.1) | 14.9 (0.8) |
| **Lauder I** (New Zealand) (−45.04, 169.68) | 99 | 3.6 (0.2) | 10.3 (0.6) | −12.3 (−0.7) | 10.4 (0.6) |
| **Lauder II** (New Zealand) (−45.04, 169.68) | 93 | 3.8 (0.2) | 11.0 (0.6) | −11.8 (−0.67) | 10.8 (0.6) |
| **Park Falls** (US) (45.94, -90.27) | 325 | −6.8 (−0.4) | 15.9 (0.9) | −29.3 (−1.6) | 17.4 (0.9) |
| **East Trout Lake** (Canada) (54.36, −104.99) | 315 | −3.7 (−0.2) | 19.0 (1.0) | −27.1 (−1.5) | 21.4 (1.2) |
| **Lamont** (US) (36.6, −97.49) | 388 | −10.1 (−0.5) | 10.0 (0.5) | −19.6 (−1.1) | 11.2 (0.6) |
| **Orléans** (France) (47.97, 2.11) | 139 | −0.07 (0.0) | 10.8 (0.6) | −16.0 (−0.9) | 12.0 (0.7) |
| **Edwards** (US) (34.95, −117.88) | 373 | −0.6 (−0.03) | 8.7 (0.5) | 7.1 (0.4) | 8.8 (0.5) |
| **Sodankylä** (Finland) (67.37, 26.63) | 234 | −10.9 (−0.6) | 19.1 (1.0) | −39.4 (−2.1) | 18.6 (1.0) |
| **Mean bias, station-to-station variability** | | −3.4 (−0.2) | 5.6 (0.3) | −17.0 (-0.9) | 11.5 (0.6) |

To filter for scenes covered with snow or ice, Wunch et al. (2011b) introduced the so-called "blended-albedo", which combines the surface albedo in the NIR and SWIR to be used as a filter. By applying it to Sodankylä and East Trout Lake, we found that a threshold value of 0.85 is optimal to remove these scenes. The influence of snow needs to be further investigated from the retrieval algorithm perspective, and it should be considered when interpreting the validation results and when analysing TROPOMI XCH$_4$ data over snow-covered scenes, most prevalent at high latitudes.

## 5  Comparison with GOSAT satellite

We compare XCH$_4$ retrieved from TROPOMI and GOSAT measurements for a period of two years (Jan 2018 – Dec 2019). The comparison yields a mean bias of −10.3 ± 16.8 ppb (−0.6 ± 0.9 %) and a Pearson's correlation coefficient of 0.85. Figure 10 shows TROPOMI and GOSAT XCH$_4$ and their ratio averaged to a 2°x 2° grid. Overall compared to GOSAT, TROPOMI





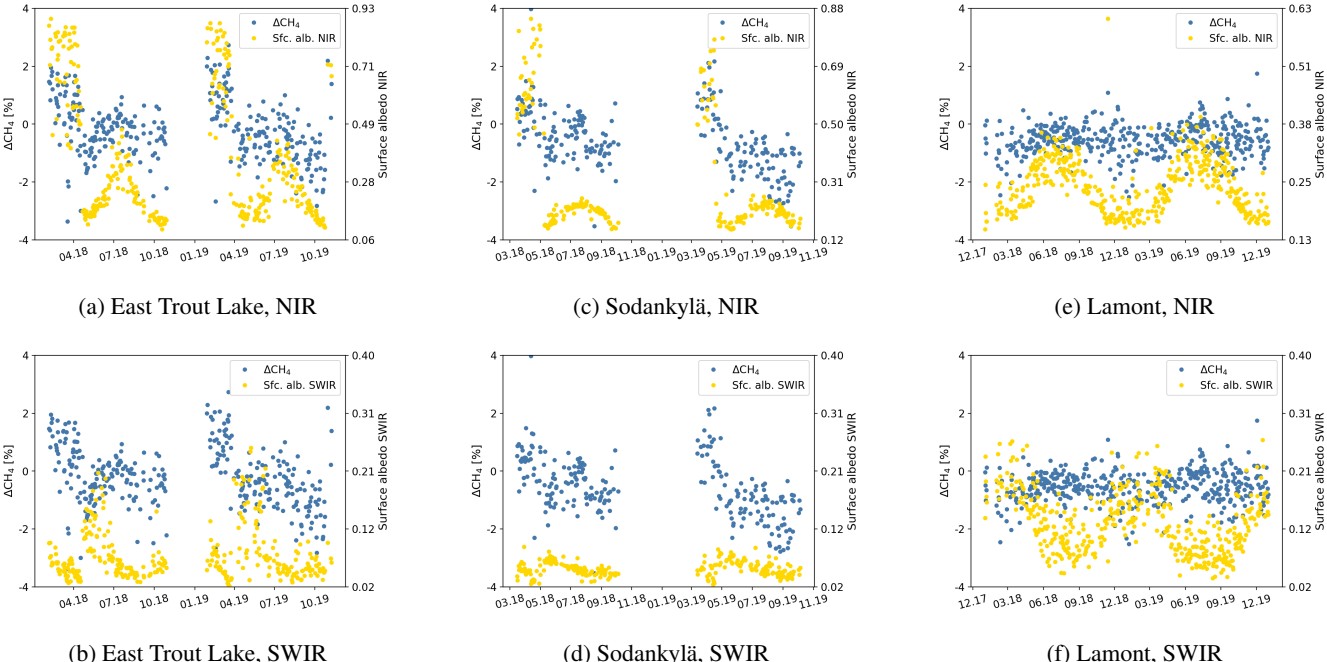

**Figure 9.** Daily mean relative differences (blue, left axis) between TROPOMI and TCCON XCH$_4$ ($\Delta$CH$_4$) and surface albedo in the NIR (yellow, secondary axis, first row) and surface albedo in the SWIR (yellow, secondary axis, second row) at East Trout Lake (54.3°N) (first column), Sodankylä (67.4°N) (second column) and Lamont (36.6°N) (third column).

underestimates XCH$_4$, specially in the regions around the tropics in South America ($-0.6 \pm 0.8$ %) and in the African continent ($-0.9 \pm 0.8$ %). In Asia there is higher variability (up to 1 %) compared to other regions, with areas of underestimation as well as overestimation. The overall underestimation is stronger by about 2 % in the non-corrected XCH$_4$, reflecting that the albedo correction improves the too low TROPOMI XCH$_4$ in areas where the surface albedo is low (e.g. forests around the Equator).

5    For higher latitudes, the underestimation is less strong, and in some areas TROPOMI overestimates XCH$_4$ compared to GOSAT (e.g. Greenland and Antarctica), in agreement with the high bias in XCH$_4$ reported in the TCCON validation at East Trout Lake and Sodankylä.

   The latitudinal distribution of XCH$_4$ from TROPOMI, GOSAT and TROPOMI collocated with TCCON stations is shown in Fig. 11, summarising the validation of TROPOMI XCH$_4$ and showing the good agreement between the three datasets. Similar

10    to Fig. 10, it shows that TROPOMI underestimates GOSAT at most latitudes but both overlap within the XCH$_4$ variability. It also shows the shift to an overestimation at high latitudes where TROPOMI retrieves higher XCH$_4$. This agrees with the conclusion that over snow TROPOMI XCH$_4$ is too high and although this distribution resembles the latitudinal distribution of XCH$_4$ shown in Fig. 1, it cannot be attributed to the selection of the spectroscopic database.





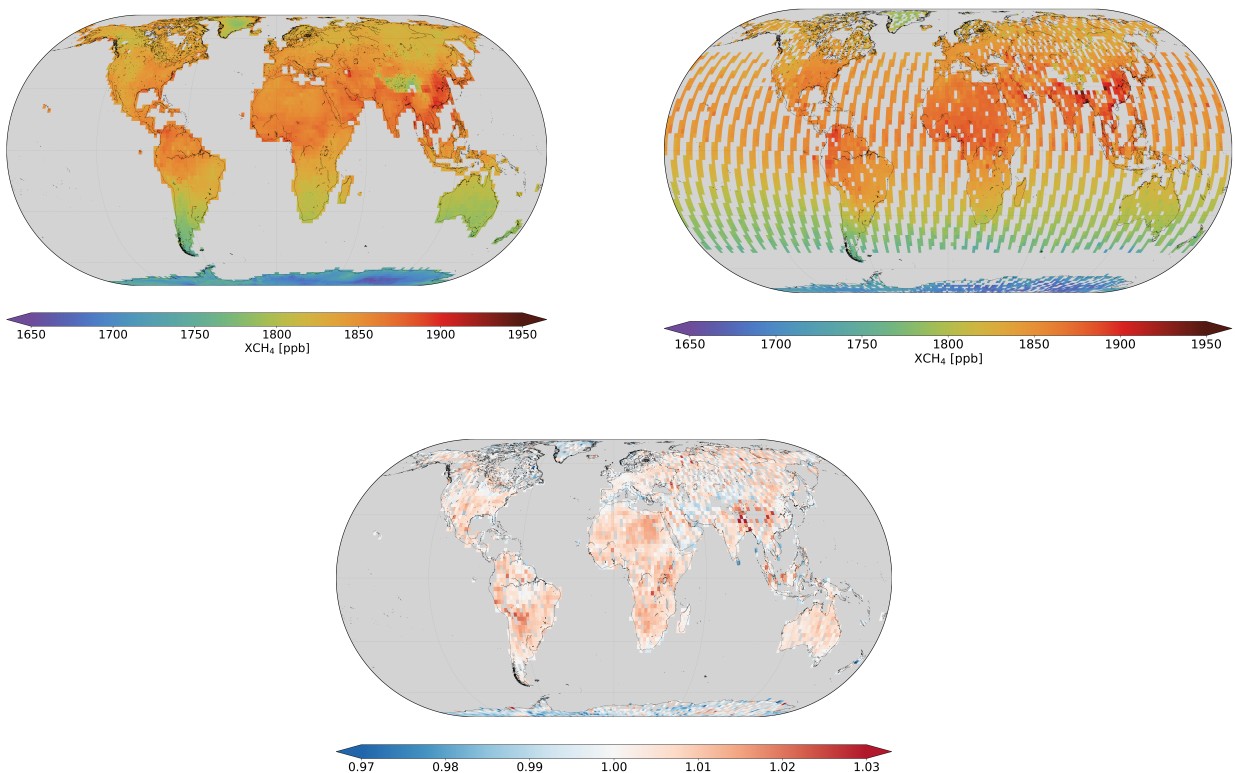

**Figure 10.** Global distribution of XCH$_4$ measured by (a) TROPOMI, (b) GOSAT and (c) the ratio of GOSAT to TROPOMI XCH$_4$. Daily collocations are averaged to a 2°x2° grid for the period 1 Jan 2018 – 31 Dec 2019.

## 6 Conclusions

We have presented several improvements that have been implemented in the retrieval of XCH$_4$ from TROPOMI measurements in the NIR and SWIR spectral range. Now that TROPOMI has been measuring for more than two years, the amount of data allows the implementation of a series of updates that were not previously possible without the use of any reference data (i.e. regularization scheme and a posteriori correction derived using only TROPOMI XCH$_4$ data).

The regularization scheme with constant regularization parameters stabilizes the retrieval and yields less scatter in the TROPOMI XCH$_4$ data compared to the operational data product (version 1.2.0 Hu et al. (2016)). We have investigated the effect of the horizontal resolution of the surface elevation database by replacing GMTED2010 S5P with the SRTM 15" database, relevant in the XCH$_4$ retrieval for which accurate knowledge of surface pressure is necessary. The higher resolution database results in a more realistic representation of surface altitude, particularly for mountainous regions and places with rough surfaces, where differences in surface pressure above 5 hPa result in retrieved XCH$_4$ that varies up to 10 ppb for specific scenes.

We have tested three state-of-the-art spectroscopic databases (HITRAN 2008 with updates from Scheepmaker et al. (2013), HITRAN 2016 and SEOM-IAS). Using the SEOM-IAS database results in the best spectral fitting quality parameters in the



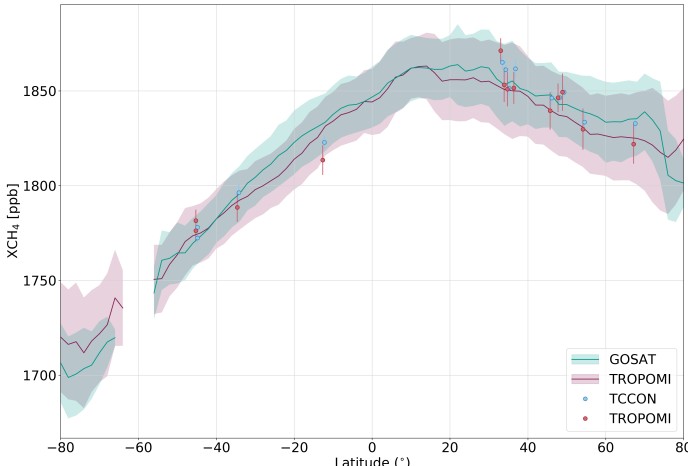

**Figure 11.** Latitudinal distribution of XCH$_4$ measured by TROPOMI and GOSAT, and the TROPOMI and TCCON collocations over the selected stations for validation in Sect. 4. The shaded bands indicate the scatter (i.e. 1$\sigma$ standard deviation) around the mean.

retrieved XCH$_4$. Each of the different spectroscopic database introduces a bias in the distribution of XCH$_4$ with respect to each other, but there is not any additional bias (e.g. latitudinal, albedo bias) that could point to the fitness for purpose of any of the databases. In view of the best fitting results, we decided to use the SEOM-IAS database, which was derived specifically for TROPOMI. However, there is a need for a thorough and detailed analysis of these databases focusing on the different absorbers

that are relevant in the CH$_4$ absorption bands to learn about the underlying processes that are driving the overall bias.

      One of the most relevant updates is the implementation of a posteriori correction that is fully independent of any reference data. We have derived a correction for the bias dependence on albedo using only TROPOMI XCH$_4$ data. This has been possible due to the high resolution of TROPOMI and its global coverage. We select regions around the globe which cover different albedo ranges and dependencies to estimate the albedo bias. The new correction is more accurate than the regular polynomial

fit for the strong XCH$_4$ underestimation at low surface albedo scenes, and also corrects for the positive bias in scenes with high surface albedo. After applying the correction, the albedo dependence in the TROPOMI-GOSAT and TROPOMI-TCCON comparison is removed, which is an independent verification of the correction scheme. The change in XCH$_4$ induced by the bias correction is in the range of 2 %, and although we attribute it mostly to unaccounted light path modification due to scattering processes, Butz et al. (2012) predicted residual scattering errors to be mostly below 1 % which suggests that other errors might

exist that needs to be further investigated.

      The good agreement of TROPOMI XCH$_4$ with TCCON ($-3.4 \pm 5.6$ ppb) and GOSAT ($-10.3 \pm 16.8$ pbb) highlights the high quality of the TROPOMI measurements. Low and high albedo scenes are the most challenging for the XCH$_4$ retrieval algorithm, and although the posteriori correction accounts for most of the bias, there is a need to further understand the





underlying cause and whether it originates in the instrument or in the retrieval algorithm. Also the overestimation of $XCH_4$ over snow covered scenes requires further investigation from the retrieval algorithm perspective. With respect to the validation, additional sites would be beneficial to cover the under-sampled regions and conditions. The network is currently limited to relatively low albedo values, so there is a lack of reference data for high albedo scenes, particularly around the Equator.

5   Furthermore, there is a clear imbalance between the number of stations in the Northern and Southern hemisphere, as well as a lack of stations below 45° S. This is not only relevant for a complete validation of current and future satellite instruments, but also to have a complete global network to monitor concentrations of $CH_4$ in the atmosphere.



## Appendix A: Filtering criteria

**Table A1.** Overview of the filters applied to assure high-quality TROPOMI XCH$_4$ retrievals.

| Parameter | Range |
|---|---|
| Cloud fraction* from VIIRS inner field of view (IFOV) | < 0.001 |
| Cloud fraction* from VIIRS outer field of view (OFOV) (upscaled FOV by 10, 50 and 100% ) (OFOVa, b, c) | < 0.001 |
| Ratio of XCH$_4$ retrieved from strong and weak absorption bands with the non-scattering retrieval using H2016 cross-sections | 0.85< x< 1.15 |
| Standard deviation of XCH$_4$ ratio within SWIR pixel plus 8 neighbouring pixels | < 0.05 |
| Signal-to-noise ratio | > 50 |
| Precision (noise-related error) | < 10 |
| $\chi^2$ | < 100 |
| Retrieved AOT (SWIR) | < 0.3 |
| Surface albedo | > 0.02 |
| Solar zenith angle (°) | < 70 |
| Viewing zenith angle (°) | < 60 |
| Terrain roughness (m) Standard deviation of surface elevation within ground pixel | < 80 |

\* Cloud fraction defined as fraction of VIIRS pixels classified as confidently clear sky.

## Appendix B: Regions selected for the posteriori correction

The regions selected to estimate the posteriori correction are shown in Fig. B1.

*Data availability.* The TROPOMI CH4 dataset of this study is available for download at ftp://ftp.sron.nl/open-access-data-2/TROPOMI/tropomi/ch4/

5 (last access: 26 June 2019). TCCON data are available from the TCCON Data Archive, hosted by CaltechDATA, California Institute of Technology, CA (US), https: //tccondata.org/ (TCCON, 2020).



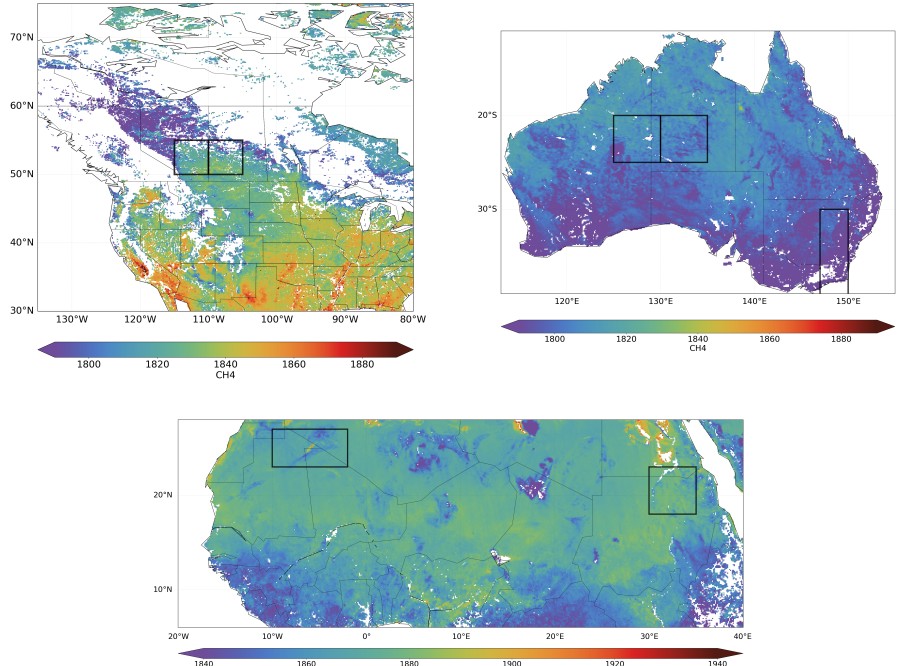

**Figure B1.** Black boxes over North Africa, Australia and Canada correspond to the different regions selected to estimate the posteriori correction (see Sect. 3.4). Global distribution of XCH$_4$ averaged to a 0.1$°$x 0.1$°$ grid for the period 1 Jan 2018 - 31 Dec 2019.

*Author contributions.* AL, TB, OH, JdB, AS, AB and JL provided the TROPOMI CH4 retrieval and data analysis. The TCCON partners provided the validation datasets. AL wrote the original draft with input from TB and JL, all authors discussed the results and review and edited the manuscript.

*Competing interests.* The authors declare that they have no conflict of interest.

5    *Disclaimer.* The presented work has been performed in the frame of Sentinel-5 Precursor Validation Team (S5PVT) or Level 1/Level 2 Product Working Group activities. Results are based on preliminary (not fully calibrated or validated) Sentinel-5 Precursor data that will still change. The results are based on S5P L1B version 1 data. Plots and data contain modified Copernicus Sentinel data, processed by SRON.

*Acknowledgements.* The TROPOMI data processing was carried out on the Dutch National e-infrastructure with the support of the SURF Co-operative. Darwin and Wollongong TCCON sites are funded by the Australian Research Council (DP140101552, DP160101598, LE0668470)
10    and NASA (NAG5-12247, NNG05-GD07G). NMD is supported by an ARC Future Fellowship FT180100327.



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
