# Peer review of "Methane retrieved from TROPOMI: improvement of the data product and validation of the first two years of measurements"

_Atmospheric Measurement Techniques, 2020_

## Referee Comment (RC1) · Anonymous Referee #1 · 19 Aug 2020

Manuscript "Methane retrieved from TROPOMI: improvement of the data product and validation of the first two years of measurements", submitted by Lorente et al. for publication in Atmos. Meas. Tech. describes retrieval algorithm improvements and related investigations carried out to generate an improved operational TROPOMI XCH4 data product in the future. The paper covers a topic relevant for Atmos. Meas. Tech. and it very well written. I recommend publication after the comments listed below have been considered by the authors.

Specific comments:

Abstract:

[Figure]

Page 1, line 2: I recommend to add "and sampling" after "spatial resolution" as TROPOMI has a similar spatial resolution as GOSAT but much denser spatial sampling.

Page 1, line 5: "The updated TROPOMI CH4 product ...": If possible, please add version number. Does this product exist, i.e., is it available for interested users? If not, then please write "The updated TROPOMI CH4 retrieval algorithm ...".

Introduction:

Page 2, line 24: Barre et al., 2020: Missing in section "References". Please add. Please add that there is (at least) one other product as described in Schneising et al., 2019, and Schneising et al., 2020. These publications need to be cited (see References below) and the results shown in Schneising et al., 2020, need to be mentioned, especially those related to the Permian basin (see line 22).

Section 2.1:

Page 4, line 15, and Eq. (4): Instrument noise is not the only contributor to "XCH4 random errors", i.e., precision, as also other instrumental (e.g., inhomogeneous scene illumination) and retrieval errors (e.g., unconsidered variability of albedo and aerosols) may contribute. I suggest to add this limitation or, alternative, state that Eq. (4) is the definition of precision as used for this manuscript.

Page 4, line 21: "In cases when VIIRS data is not available, we use a back-up ...": Does this happen? If yes, I would expect that this results in inconsistencies. Please add more information.

Page 4, line 27 following: "This updated retrieval algorithm is referred to as the beta-version of the TROPOMI XCH4 data product." Sentence not OK. An algorithm is not a data product.

Section 2.2:

Below Tab. 1: "*For the Lauder station the ll instrument was replaced on October 2018 to ll.". ll replaced by ll?

Page 5, line 9: If the TROPOMI data are averaged daily then I assume that the TROPOMI XCH4 averaging kernels have not been considered for the validation. Please add more info on this aspect.

Section 2.3:

Page 6, line 17: "both retrievals performed similarly": With respect to what? Likely not w.r.t. yield as number of data points in proxy product is much higher. Please refine the statement.

Section 3.1:

Page 7, line 4: "and that retrieved aerosol parameters have realistic distributions". This is a strong (but unproven) statement. It needs to be shown in this paper that this is true.

Page 7, line 12: "19.7 ppb to 24.5 ppb": What does this mean? Is it a min to max range?

Section 3.2:

Concerning: Page 8, 6-7: "we have decided to use the SEOM-IAS spectroscopy database." I am not convinced. Was this a "political" decision? I conclude from Tab. 2 that HITRAN 2008 (used so far) is better. Is a slightly better fit quality (which can have many reasons in addition to spectroscopy) really a good argument if bias and scatter are getting larger?

Section 3.4:

Is this bias correction for albedo really new? As far as I know, the current operational XCH4 product already offers a bias corrected product. Please clarify.

Section 3.4:

Page 12, line 3: surface albedo "As": Is this the SWIR albedo? How is the NIR albedo considered?

Section 4.2:

Tab. 3: Add explanation for numbers in brackets. Is this 1-sigma uncertainty?

Typos etc.:

Page 22, line 4; page 23, . . ., and possibly other places: CH4: The number 4 must be set low.

References:

Schneising, O., Buchwitz, M., Reuter, M., Bovensmann, H., Burrows, J. P., Borsdorff, T., Deutscher, N. M., Feist, D. G., Griffith, D. W. T., Hase, F., Hermans, C., Iraci, L. T., Kivi, R., Landgraf, J., Morino, I., Notholt, J., Petri, C., Pollard, D. F., Roche, S., Shiomi, K., Strong, K., Sussmann, R., Velazco, V. A., Warneke, T., and Wunch, D.: A scientific algorithm to simultaneously retrieve carbon monoxide and methane from TROPOMI onboard Sentinel-5 Precursor, Atmos. Meas. Tech., 12, 6771-6802, https://doi.org/10.5194/amt-12-6771-2019, https://doi.org/10.5194/amt-12-6771-2019, 2019.

Schneising, O., Buchwitz, M., Reuter, M., Vanselow, S., Bovensmann, H., and Burrows, J. P.: Remote sensing of methane leakage from natural gas and petroleum systems revisited, Atmos. Chem. Phys., 20, 9169-9182, https://doi.org/10.5194/acp-20-9169-2020, 2020.
* * *

---

## Referee Comment (RC2) · Anonymous Referee #2 · 8 Oct 2020

Review of "Methane retrieved from TROPOMI: improvement of the data product and validation of the first two years of measurements" – Alba Lorente et al.

General comments I found this paper to be of good scientific significance since it details the progress of the RemoTeC full-physics algorithm for TROPOMI data. This includes the use of a new bias correction routine to account for albedo biases which is independent of other sources of validation data; as well as the evaluation of spectroscopic databases and comparisons of the data to both TCCON and GOSAT data. The authors also evaluate biases which they see at higher latitudes over potentially snowy scenes. The new bias corrected TROPOMI data that they present is a good improvement over

the previous data without the correction. The scientific quality of the data is good, using valid methods and approaches that are discussed and analysed appropriately and with appropriate references. The presentation is very good, the English is very good, the figures and tables are clear and understandable. Overall I recommend the manuscript to be accepted for publication in AMT after the authors have address my comments below.

Specific comments 1. Page 4, line 5. How do you determine the position of the 12 pressure layers? Are they fixed for every scene or are they calculated with respect to the surface pressure or tropopause height? If the location of the tropopause isn't accounted for in the construction of vertical layers, have you considered the uncertainty this could cause in calculating the total column XCH4, compared to a method which aims to put a pressure layer boundary at the tropopause height?

2. Page 4, line 12. Could you please be more specific in which ECMWF data you are using. Is it ERA-5 or IRA-Interim for example.

3. Page 4, line 21. How well do the results of the back-up filter compare to the VIIRS cloud filter data when you try to use it for scenes where you do have VIIRS to validate it? How many scenes in total for your two years of data use the VIIRS cloud clearing method, and how many use the back-up H2O retrieval method?

4. Page 4, lines 24-26. I am a little confused by how you cite a paper from 2019 (Hasekamp et al. 2019) to say that results of version 1.2.0 from June 2020 of your algorithm largely comply with mission requirements. Please could you elaborate on this.

5. Page 4, line 27. You call this new version the beta version here, but do not refer to this again. However, on page 6, line 20 you say the updates to the algorithm correspond to v1.3.0. Is there a difference between this beta version and 1.3.0? If not then it might be clearer to call it v 1.3.0 here on page 4.

6. Page 5, line 8. Could you please comment on why you chose to use daily averaged TCCON instead of averaging only data which is within a shorter time frame. I understand that TROPOMI has 14 orbits in one day so I would assume it likely that more than one orbit may intersect the 600km diameter co-location criteria. Do you think there is merit in being stricter in your temporal co-location as a result so you are only matching TCCON at a similar time to an overpass?

7. Page 6, line 4. I think it's potentially misleading to say that the GOSAT swath is 790km with a 10.5km resolution without saying that its measurement method is different to TROPOMI's and that it usually only makes 3 of those 10.5km measurements across its swath. I think an additional sentence here on the sampling pattern of GOSAT would be helpful.

8. Page 6, line 16. You use a full-physics method for TROPOMI because the proxy method cannot be applied, and go on to say that the full-physics and proxy methods were found to perform similarly for GOSAT. What you don't explain in the paper is why you don't use the GOSAT full-physics data as this feels like a more natural comparison. Could you please comment on why you used gosat proxy over gosat full-physics?

9. On Section 3.1. It makes sense to me that using a constant gamma reduces the overall dispersion of the data, improving your results. But please can you comment on the theory behind why you think calculating gamma per iteration should result in a less accurate result than using an average value.

10. Page 7, line 12. The reduction of 9% going from 19.7 ppb to 24.5 ppb doesn't make sense to me since it's becoming larger instead of reducing and the difference between these numbers is larger than 9%.

11. Page 7, line 25. In your section on the TCCON validation you say that the overall bias with respect to HITRAN 2008 is +15.5 for HITRAN 2016. Table 2 shows that the difference between HITRAN 2008 and HITRAN 2016 is 20.3 ppb for TCCON.

12. Page 7, line 31. Please could you give the global numbers as referred to here which show that SEOM-IAS has a significantly improved RMS and chi-squared over the other two.

13. On section 3.3. I like the discussion on the differences of greater than 45m and 50m, but in figure 2 there are a lot of smaller systematic differences of 10-20m to be seen in the Eastern US which lead to a net positive XCH4 difference over this half of the country. Firstly, please could you comment on why you think the higher resolution DEM would be on average higher elevation than the lower-res DEM over this region. And following on, could you please comment on the change to XCH4 overall as a result of any mean altitude difference between DEMs on a global scale (if one exists). I ask since you only focus on outliers between the DEMs in the paper and don't talk about any systematic differences.

14. On the Small Area Analysis. Could you please elaborate on how and why you chose the areas which you did. How dependent on your method is the choice of SAAs.

15. On the bias correction method, page 10, line 1. You only apply a bias correction on the surface albedo and say that the other retrieved parameters show negligible dependence, showing surface albedo, AOD and SZA. For OCO-2 the parameter dP (the difference between the a priori and retrieved surface pressure) shows the largest dependence on the bias. Is this a parameter you have looked in to?

16. Page 16. Lines 5-6. Have you tried comparing with snow cover data to verify how reliable this method of detecting snow actually is?

Technical corrections âǍć Page 5, Table 1. Caption missing versions for the instrument. âǍć Page 10, line 7. Typo with full stop. I assume you wanted a capital T or a semi-colon. âǍć There are multiple instances throughout the paper where you've misspelt ppb as pbb. Such as Page 7 line 25, page 9 line 8, page 20 line 16 and twice on page 14 line 4.

---

## Author Response (AR1)

**Response to the reviewers on the manuscript "Methane retrieved from TROPOMI: improvement of the data product and validation of the first two years of measurements" by Alba Lorente et al.**

The authors would like to thank the reviewers for their thoughtful and helpful comments and suggestions. Below are the comments by the reviewers in blue and replies in black. Any modification made to the text has been underlined. The line and page numbers correspond to the version of the manuscript available for online discussion.

**Reviewer 1**

**Comment C 1.1** — Page 1, line 2: I recommend to add "and sampling" after "spatial resolution" as TROPOMI has a similar spatial resolution as GOSAT but much denser spatial sampling.

**Reply**: Added. We have also been more specific on the sampling technique of GOSAT on page 6, line 4.

**Comment C 1.2** — Page 1, line 5: "The updated TROPOMI CH4 product...": If possible, please add version number. Does this product exist, i.e., is it available for interested users? If not, then please write "The updated TROPOMI CH4 retrieval algorithm...".

**Reply**: It is an existing product and it is publicly available through the ftp specified in the "Data availability" section. However, as we do not want to confuse the reader with version numbers and detailed specifications about the product in the abstract, we have modified the sentence to "The updated retrieval algorithm..." as the features that follow in that sentence refer to the algorithm itself.

**Comment C 1.3** — Page 2, line 24: Barre et al., 2020: Missing in section "References". Please add. Please add that there is (at least) one other product as described in Schneising et al., 2019, and Schneising et al., 2020. These publications need to be cited (see References below) and the results shown in Schneising et al., 2020, need to be mentioned, especially those related to the Permian basin (see line 22).

**Reply**: We have added Barre et al. (2020) to the reference list; this was forgotten because when preparing this manuscript it was still under discussion in ACPD.

We agree that the WFM-DOAS TROPOMI product (Schneising et al., 2019) should be mentioned. We have mentioned it in Section 2.1 (TROPOMI CH$_4$ retrieval algorithm), page 4, line 27. We think this location fits better as it is here where the retrieval algorithm is presented. The added

text is: "Another scientific retrieval algorithm using the Weighting Function Modified Differential Optical Absorption Spectroscopy (WFM-DOAS) method to retrieve CO and CH$_4$ from TROPOMI was presented by Schneising et al. (2019). Comparison of both retrieval approaches is foreseen as part of ongoing verification activities."

We have added Schneising et al. (2020) when we refer in the text to the studies of the Permian basin. We do not go into the details of neither Schneising et al. (2020) nor Zhang et al. (2020) as the aim of this paragraph is to highlight some of the studies that have successfully used TROPOMI XCH$_4$ data to derive emissions.

**Comment C 1.4** — Page 4, line 15, and Eq. (4): Instrument noise is not the only contributor to "XCH4 random errors", i.e., precision, as also other instrumental (e.g., inhomogeneous scene illumination) and retrieval errors (e.g., unconsidered variability of albedo and aerosols) may contribute. I suggest to add this limitation or, alternative, state that Eq. (4) is the definition of precision as used for this manuscript.

We acknowledge that there are other contributions to the random error besides the measurement noise. So it is true that Eq. 4 is the definition of the precision given in the product and so as used for this manuscript. As suggested by the reviewer, we explicitly mention this. "The precision $\sigma_{\mathrm{XCH}_4}$ available in the data product is defined as the standard deviation of the retrieval noise".

**Comment C 1.5** — Page 4, line 21: "In cases when VIIRS data is not available, we use a back-up...": Does this happen? If yes, I would expect that this results in inconsistencies. Please add more information.

**Reply**: Data from VIIRS is hardly ever not available, so this does not happen very often. VIIRS data used in the TROPOMI XCH$_4$ retrieval is processed operationally by the S5P-NPP cloud processor. If due to any circumstance the processing of the VIIRS data fails or it is delayed, we use this filtering as a back up option. The XCH$_4$ data is flagged accordingly (qa value downgraded to 0.4) to avoid any possible inconsistencies as mentioned by the reviewer. From all the orbits processed operationally since the beginning of the mission, for less than 1% the processing of VIIRS data was not nominal in the CH$_4$ retrieval.

We added the following to clarify this point: "In less than 1% of the cases when VIIRS data is not available, we use a back-up filter based on a non-scattering CH$_4$ retrieval from the weak and strong absorption bands (Hu et al., 2016). These cases are flagged accordingly by the quality value indicator."

**Comment C 1.6** — Page 4, line 27 following: "This updated retrieval algorithm is referred to as the beta version of the TROPOMI XCH4 data product." Sentence not OK. An algorithm is not a data product.

**Reply**: We agree with the reviewer about the misleading terminology used here. We have modified the text as follows in page 4, line 27: "The TROPOMI XCH$_4$ scientific data product from SRON retrieved with the updated algorithm serves as a beta version of the operational processing."

Following this comment, we have further clarified at the beginning of Sect. 3 (page 6, line 20), removing the reference to version 1.3.0 that will eventually correspond to the future operational update but this is not certain as of now: "The TROPOMI XCH$_4$ scientific data product from SRON retrieved with the updated algorithm will be suggested for use in the operational processing in the next processor update.."

**Comment C 1.7** — Below Tab. 1: "*For the Lauder station the ll instrument was replaced on October 2018 to ll.". ll replaced by ll?

**Reply**: We thank the referee for spotting the typo. The instrument "ll" (Sherlock et al., 2017) was replaced by "lr" (Pollard et al., 2019). We have corrected this.

**Comment C 1.8** — Page 5, line 9: If the TROPOMI data are averaged daily then I assume that the TROPOMI XCH$_4$ averaging kernels have not been considered for the validation. Please add more info on this aspect.

**Reply**: The total column averaging kernel can only be used when CH$_4$ profile measurements with a high vertical resolution would be available for validation. However, the measurements from the TCCON network only provide total column integrated measurements which hampers the application of the averaging kernels.

**Comment C 1.9** — Page 6, line 17: "both retrievals performed similarly": With respect to what? Likely not w.r.t. yield as number of data points in proxy product is much higher. Please refine the statement.

**Reply**: We agree with the reviewer that we should be more specific in this statement. We modify the text for that purpose: "...both retrievals performed similarly with respect to bias variability and precision when validating the retrieved XCH$_4$ with ground-based TCCON measurements. This study also concluded that both methods can retrieve XCH$_4$ in aerosol loaded scenes with retrieval errors of less than 1%."

**Comment C 1.10** — Page 7, line 4: "and that retrieved aerosol parameters have realistic distributions". This is a strong (but unproven) statement. It needs to be shown in this paper that this is true.

**Reply**: We agree with the reviewer that this statement needs clarification. First, to avoid mis-interpretation of the output of the retrieval to which we refer as "retrieved aerosols parameters",

we change the reference to them in the manuscript to "scattering parameters" instead of "aerosol parameters", and add the prefix *effective* ("effective aerosol distribution height", "effective size parameter" and "effective aerosol column"). With effective we want to highlight that these retrieved parameters are auxiliary parameters that characterize the scattering properties of the atmosphere in the radiative transfer model in the retrieval for which the target is XCH$_4$. The aerosol parameters are only effective ones but follow a distribution that we would expect, and that is what we meant by realistic distributions. We have modified the sentence in page 7, line 4:"[...] retrieved scattering parameters follow a distribution that we would expect".

**Comment C 1.11** — Page 7, line 12: "19.7 ppb to 24.5 ppb": What does this mean? Is it a min to max range?

**Reply**: It refers to the reduction on the standard deviation of the differences mentioned at the beginning of the sentence. We add 'from', and correct the order because the reduction is from 24.5 to 19.7 ppb. Furthermore, there was a typo and 24.5 ppb is 21.5 ppb, which matches the 9% reduction specified in that same sentence.

**Comment C 1.12** — Page 8, 6-7: "we have decided to use the SEOM-IAS spectroscopy database." I am not convinced. Was this a "political" decision? I conclude from Tab. 2 that HITRAN 2008 (used so far) is better. Is a slightly better fit quality (which can have many reasons in addition to spectroscopy) really a good argument if bias and scatter are getting larger?

**Reply**: We acknowledge that the text can be somewhat misleading. The "slightly" better fit quality refers to the results when looking only to retrievals around the TCCON stations. On a global scale (page 7, line 30) "we see that both the RMS and $\chi^2$ improve significantly when using the SEOM-IAS database, with HITRAN 2008 giving the worst fitting results". The prove of this statement is not visually shown in the manuscript, but we have added the following to the text as suggested by Referee # 2 (comment 2.12): "Global mean $\chi^2$ improves by 19% with SEOM-IAS cross-section and by 7% with HITRAN 2016 with respect to HITRAN 2008."

Figure R1 below shows the ratio of $\chi^2$ of the retrieval with HITRAN 2008 and HITRAN 2016 (left) and HITRAN 2008 and SEOM-IAS (right), for one year of data averaged into daily 1° x 1° grid, which shows that SEOM-IAS cross section results in a significantly better $\chi^2$ with respect to HITRAN 2008 and HITRAN 2016. In the sensitivity tests, the only parameter that changed in the retrieval was the spectroscopic database, so any difference in the retrieval results could be attributed to the different spectroscopy. From this we concluded (page 8, line 6) "In view of the better spectral fitting results in the retrieved XCH$_4$ we have decided to use the SEOM-IAS spectroscopy database".

Regarding the results shown in Table 2, it shows that each of the spectroscopic databases introduces an overall bias that cannot be used as an independent argument to favour a specific

database, as the comparison to GOSAT and TCCON might also be biased because of the specific spectroscopy used in their retrievals. The variation in the scatter of 1 to 3 ppb is not conclusive, as this is negligible if compared to the magnitude of other sensitivities and errors in the retrieval.

[Figure]

Figure R1: Ratio of $\chi^2$ from the retrieval with HITRAN 2008 and HITRAN 2016 (left) and SEOM-IAS (right).

**Comment C 1.13** — Section 3.4. Is this bias correction for albedo really new? As far as I know, the current operational $XCH_4$ product already offers a bias corrected product. Please clarify.

**Reply**: Indeed, the operational $XCH_4$ product already has a posteriori correction applied to it. The novelty of the bias correction presented in this study is the way we have derived it, as we have not used any external or reference data (like GOSAT or TCCON) to estimate the dependence, and the fit to the dependence on surface albedo is done differently.

The new approach is explained in page 10, line 10 – page 11, line 3. Also in page 11, line 19 we refer to the approach in the operational compared to the new fit: "for which the B-spline fit corrects more strongly than the regular polynomial fit that was previously used."

We try to make it clearer by modifying the text:

- Page 10, line 10: "In the baseline operational algorithm few months after TROPOMI was operational, we applied a correction..."

- Page 10, line 12: "[..] we have sufficient data to derive a new  correction".

**Comment C 1.14** — Page 12, line 3: surface albedo "$A_s$": Is this the SWIR albedo? How is the NIR albedo considered?

**Reply**: In the correction we only consider the surface albedo in the SWIR spectral range, as the dependence of the bias on the surface albedo in the NIR spectral range (see Fig. R2) is negligible compared to the dependence shown in Fig. 3a for the surface albedo in the SWIR.

For clarification, we specify after Eq. 6 that $A_s$ refers to the surface albedo in the SWIR, and in page 10, line 2: "The comparison of TROPOMI [...] shows a dependence of the bias [...] on surface albedo retrieved in the SWIR spectral range".

[Figure]

Figure R2: Ratio of $XCH_4$ measurements by TCCON and TROPOMI as a function of retrieved surface albedo in the NIR spectral range, to compare with Fig. 3a in the manuscript.

**Comment C 1.15** — Tab. 3: Add explanation for numbers in brackets. Is this 1-sigma uncertainty?

**Reply**: The number in parenthesis are the percentage number. We have added to the caption of Table 3: "The table shows [...] (in ppb and in percentages between parenthesis)."

**Comment C 1.16** — Typo in CH4 in several places.

**Reply**: Thank you for spotting this. Changed CH4 to $CH_4$

**References**

Barré, J., Aben, I., Agustí-Panareda, A., Balsamo, G., Bousserez, N., Dueben, P., Engelen, R., Inness, A., Lorente, A., McNorton, J., Peuch, V.-H., Radnoti, G., and Ribas, R.: Systematic detection of local CH4 emissions anomalies combining satellite measurements and high-resolution forecasts, Atmos. Chem. Phys. Discuss., https://doi.org/10.5194/acp-2020-550, in review, 2020.

**Response to the reviewers on the manuscript "Methane retrieved from TROPOMI: improvement of the data product and validation of the first two years of measurements" by Alba Lorente et al.**

The authors would like to thank the reviewers for their thoughtful and helpful comments and suggestions. Below are the comments by the reviewers in blue and replies in black. Any modification made to the text has been underlined. The line and page numbers correspond to the version of the manuscript available for online discussion.

**Reviewer 2**

**Comment  C 2.1**  —  Page 4, line 5. How do you determine the position of the 12 pressure layers? Are they fixed for every scene or are they calculated with respect to the surface pressure or tropopause height? If the location of the tropopause isn't accounted for in the construction of vertical layers, have you considered the uncertainty this could cause in calculating the total column $XCH_4$, compared to a method which aims to put a pressure layer boundary at the tropopause height?

**Reply**:   The equidistant pressure layers are determined by the surface pressure and the top of atmosphere from the meteorological input, constrained by a value of 0.1 hPa. So the grid differs per retrieval, and during a single retrieval it remains fixed as the algorithm does not retrieve surface pressure. For the a priori vertical profile we use TM5 that varies with latitude, longitude and altitude also accounting for the effect of tropopause height variations.

**Comment  C 2.2**  —  Page 4, line 12. Could you please be more specific in which ECMWF data you are using. Is it ERA-5 or ERA-Interim for example.

**Reply**:  The ECMWF data that we use is an operational analysis product; it corresponds to the first analysis performed after the forecast product, so it is not a reanalysis product as ERA-5 or ERA-Interim. Access to this ECMWF product is granted to us on behalf of the TROPOMI project. We have added "operational analysis product" to the text to make it clearer.

**Comment  C 2.3**  —  Page 4, line 21. How well do the results of the back-up filter compare to the VIIRS cloud filter data when you try to use it for scenes where you do have VIIRS to validate it? How many scenes in total for your two years of data use the VIIRS cloud clearing method, and how many use the back-up H2O retrieval method?

**Reply**:  The cloud filtering is important in our processing, so we acknowledge the referee bringing it up, and we hope to clarify it as the Referee #1 also raised a question on this topic.

The quality of our XCH$_4$ retrieval relies on a very strict cloud filtering, for which we use VIIRS data that is able to identify small-scale cloud structures that could lead to errors in the retrieval if not filtered properly. VIIRS data used in the TROPOMI XCH$_4$ retrieval is processed operationally by the S5P-NPP cloud processor. If due to any circumstance the processing of the VIIRS data fails or it is delayed, we use the filtering based on a non-scattering retrieval as a back-up option. In this circumstance, the XCH$_4$ data is flagged (qa value downgraded to 0.4) as the data might be affected by cloud contamination, because the non-scattering retrieval is not as effective filter as VIIRS data, particularly for thick clouds. Originally, the filter based on the the non-scattering retrieval was optimized to filter cirrus over dark surfaces (Hasekamp et al., 2019), by applying a 6% and 22% threshold for the difference between the CH$_4$ and H$_2$O retrieved in the weak and strong absorption bands. This filter alone will effectively remove scenes with a cloud fraction higher than 15%, a fraction that is too high to keep the errors in the retrieved CH$_4$ below requirements. Together with the scattering filter (using the retrieved scattering parameters) scenes with a cloud fraction higher than 8% will be effectively filtered, but still far from the desired 1-2% for the CH$_4$ retrieval (Hasekamp et al., 2019). These numbers presented here correspond to the analysis made prior to launch, that need to be repeated using real data.

As VIIRS data is operationally processed, it is rarely missing or not available for its use in the XCH$_4$ retrieval. From all the orbits processed operationally since the beginning of the mission, for less than 1% the processing of VIIRS data was not nominal in the CH$_4$ retrieval. We added the following to stress this point: "In less than 1% of the cases when VIIRS data is not available, we use a back-up filter based on a non-scattering H$_2$O and CH$_4$ retrieval from the weak and strong absorption bands (Hu et al., 2016). These cases are flagged accordingly by the quality value indicator."

**Comment C 2.4** — Page 4, lines 24-26. I am a little confused by how you cite a paper from 2019 (Hasekamp et al. 2019) to say that results of version 1.2.0 from June 2020 of your algorithm largely comply with mission requirements. Please could you elaborate on this.

**Reply**:

Hasekamp et al. (2019) is the reference to the ATBD for the operational algorithm version 1.2.0 mentioned in that sentence. We agree with the reviewer that this might be confusing for the reader, so we remove "as of June 2020". We wanted to specify the version of the operational algorithm when the manuscript was written/submitted (and that is why we added "as of June 2020"), having in mind that this version could have changed in the meantime. But we acknowledge that with the version number it should be sufficient to trace it back.

**Comment C 2.5** — Page 4, line 27. You call this new version the beta version here, but do not refer to this again. However, on page 6, line 20 you say the updates to the algorithm correspond

to v1.3.0. Is there a difference between this beta version and 1.3.0? If not then it might be clearer to call it v 1.3.0 here on page 4.

**Reply**: We acknowledge that the naming and version numbers might led to confusion when reading it, a remark also made by Referee #1 in comment C1.6. We try to make it clearer through the manuscript.

The reference to *beta version of the TROPOMI XCH$_4$ data product* is used for the data product that results from the scientific development activities within the L2 team at SRON. The next step for these developments is to be implemented in the operational processing whenever there is a a processor update.

We have removed the reference to version 1.3.0 that will eventually correspond to the future operational update, because this specific numbering is not certain as of now, and only causes confusion. Now page 6, line 20 reads: "The TROPOMI XCH$_4$ scientific data product from SRON retrieved with the updated algorithm will be suggested for use in the operational processing in the next processor update."

**Comment C 2.6** — Page 5, line 8. Could you please comment on why you chose to use daily averaged TCCON instead of averaging only data which is within a shorter time frame. I understand that TROPOMI has 14 orbits in one day so I would assume it likely that more than one orbit may intersect the 600km diameter co-location criteria. Do you think there is merit in being stricter in your temporal co-location as a result so you are only matching TCCON at a similar time to an overpass?

**Reply**: We are glad that this was brought up as we do limit the TCCON measurements to ± 2 hours of the TROPOMI overpass, so the explanation on the manuscript is wrong, and we have changed it accordingly.

The mistake on the text is because we performed sensitivity tests by also using daily averages. Figure R1 shows the validation results with time constraint (left, same as Fig. 8a in the original manuscript) and without any time constraint (right). The overall validation results do not change significantly. The mean bias does not change significantly, and the station to station variability is only affected by 1 ppb. The number of collocation pairs did increase significantly (from 3203 to 8351). As an example for the validation over one of the stations, Fig. R2 shows the time series of the bias with time constraint (left) and without any time constraint (right).

**Comment C 2.7** — Page 6, line 4. I think it's potentially misleading to say that the GOSAT swath is 790km with a 10.5km resolution without saying that its measurement method is different to TROPOMI's and that it usually only makes 3 of those 10.5km measurements across its swath. I think an additional sentence here on the sampling pattern of GOSAT would be helpful.

[Figure]

Figure R1: Mean differences between TROPOMI and TCCON XCH$_4$ ($\Delta$XCH$_4$) and the standard deviation of the differences ($\sigma_{\text{XCH}_4}$) with (left) $\pm$ 2 hours of time constraint in TCCON (as in the manuscript) and (right) daily averages.

**Reply**: We have modified the sentence in page 6, line 4: "GOSAT was launched in 2009, and  it performs three point observations in a cross-track swath of 790 km with 10.5 km resolution on the ground at nadir, which results in global coverage  approximately every 3 days".

**Comment   C 2.8** — Page 6, line 16. You use a full-physics method for TROPOMI because the proxy method cannot be applied, and go on to say that the full-physics and proxy methods were found to perform similarly for GOSAT. What you don't explain in the paper is why you don't use the GOSAT full-physics data as this feels like a more natural comparison. Could you please comment on why you used gosat proxy over gosat full-physics?

**Reply**: The main reason to use the proxy product in this comparison is the fact that the data yield is higher. Furthermore, the comparison of TROPOMI XCH$_4$ and GOSAT with both approaches results in similar bias: mean bias of $-10.3 \pm 16.8$ ppb and a Pearson's correlation coefficient of 0.85 with the proxy approach (as stated in the manuscript) and mean bias of $-12.5 \pm 14.9$ ppb and

[Figure]

Figure R2: Time series of the bias between TROPOMI and TCCON XCH$_4$ over the Lamont station with time constraint (left) and without time constraint (right).

a Pearson's correlation coefficient of 0.86 with the full physics approach. The correlation plot of both comparisons is shown in Fig. R3, on the left for GOSAT proxy and on the right for GOSAT full-physics.

We have added this information to make clear the reason for the selection of the full-physics approach (page 6, line 15): "In the validation in Sect. 5 we found that there is no bias between the GOSAT proxy and full-physics products. However, we have selected for the comparison the GOSAT proxy product over the full-physics because of its higher data yield". And we also include the results for the full-physics in a sentence in page 17, line 7: "The overall comparison yields a mean bias of $-12.5 \pm 14.9$ ppb if we use the GOSAT XCH$_4$ product retrieved with the full-physics approach".

[Figure]

[Figure]

Figure R3: Correlation plot of TROPOMI XCH$_4$ and GOSAT XCH$_4$ retrieved with the proxy approach (left) and with the full physics (right). Daily collocations are averaged to a $2° \times 2°$ grid for the period 1 Jan 2018 – 31 Dec 2019.

**Comment C 2.9** — On Section 3.1. It makes sense to me that using a constant gamma reduces the overall dispersion of the data, improving your results. But please can you comment on the theory behind why you think calculating gamma per iteration should result in a less accurate result than using an average value.

**Reply**: In theory calculating a gamma for each iteration should be actually superior than using an average gamma value. However, it is based on the idea of finding the minimum value of the elbow plots ($\chi^2$ vs. regularization strength). Using real data, such a minimum does not exist in most cases and therefore can result in a more unstable inversion. We found using an average value for gamma results in a more stable retrieval and reduces the overall dispersion of the data.

**Comment C 2.10** — Page 7, line 12. The reduction of 9% going from 19.7 ppb to 24.5 ppb doesn't make sense to me since it's becoming larger instead of reducing and the difference between these numbers is larger than 9%.

**Reply**: We appreciate the careful reading of the referee that led to spotting this typo. The reduction is from 21.5 ppb (and not 24.5) to 19.7 ppb, which corresponds to approximately 9%. We have corrected this.

**Comment C 2.11** — Page 7, line 25. In your section on the TCCON validation you say that the overall bias with respect to HITRAN 2008 is +15.5 for HITRAN 2016. Table 2 shows that the difference between HITRAN 2008 and HITRAN 2016 is 20.3 ppb for TCCON.

**Reply**: We thank again for this careful check of the numbers. Indeed the bias from -2.4 ppb to 17.9 ppb is 20.3 and not 15.5 as it is written in the text (this is considering HITRAN 2008 bias as 2.4 ppb and not -2.4 ppb). We have corrected the text accordingly.

**Comment C 2.12** — Page 7, line 31. Please could you give the global numbers as referred to here which show that SEOM-IAS has a significantly improved RMS and chi-squared over the other two.

**Reply**: We added these numbers to the text, page 7, line 31. "Global mean $\chi^2$ improves by 19% with SEOM-IAS cross-section and by 7% with HITRAN 2016 with respect to HITRAN 2008".

Figure R4 shows the ratio of $\chi^2$ of the retrieval with HITRAN 2008 and HITRAN 2016 (left) and HITRAN 2008 and SEOM-IAS (right), for one year of data averaged into daily 1° x 1° grid, which shows that SEOM-IAS cross section results in a significantly better $\chi^2$ with respect to HITRAN 2008 and HITRAN 2016.

**Comment C 2.13** — On section 3.3. I like the discussion on the differences of greater than 45m and 50m, but in figure 2 there are a lot of smaller systematic differences of 10-20m to be seen in

[Figure]

Figure R4: Ratio of $\chi^2$ from the retrieval with HITRAN 2008 and HITRAN 2016 (left) and SEOM-IAS (right).

the Eastern US which lead to a net positive XCH4 difference over this half of the country. Firstly, please could you comment on why you think the higher resolution DEM would be on average higher elevation than the lower-res DEM over this region. And following on, could you please comment on the change to XCH4 overall as a result of any mean altitude difference between DEMs on a global scale (if one exists). I ask since you only focus on outliers between the DEMs in the paper and don't talk about any systematic differences.

**Reply**: The different East-West features over United States shown in Fig. 2 are only present in this region. As why on average the SRTM results higher in elevation over Eastern US we are not sure, but we assume that compared to the S5P-DEM, the SRTM database is a better representation on the terrain over the US as it is a database that uses national data and models. Globally, we see similar features as in the Western US in most of the mountain regions around the world, so there are not systematic differences. So overall XCH$_4$ changes are more pronounced over mountain regions, that is why we focused on the outliers in the discussion on Sect. 3.3.

**Comment  C 2.14** — On the Small Area Analysis. Could you please elaborate on how and why you chose the areas which you did. How dependent on your method is the choice of SAAs.

**Reply**: The reasoning for the choice of the specific areas used in the SAA analysis was mainly to have a representation of the challenging scenes for the XCH$_4$ retrieval, mainly low and high albedo. The areas also needed to include scenes with surface albedo around the reference value, and not include (as much as possible) big sources of methane, although this was less of a limiting factor because XCH$_4$ distribution is normalized for each region separately. Furthermore, we aimed at areas that had a relatively good coverage through all the different seasons, and we stayed away from big mountain regions.

For high albedo scenes it was straightforward to chose Sahara desert, and over this area we

tested the choice of multiple regions and their size. The main challenge was to find regions that included the surface albedo reference value, and all the areas that we tested resulted in similar $XCH_4$ dependencies. Over Australia we made boxes of $5°$x $5°$ and discarded those that had in the same box different modes in the $XCH_4$ distribution with respect to surface albedo. The most southern box was interesting because it includes low surface albedo values with strong $XCH_4$ underestimation, and the shape of this area is different to $5°$x$5°$ to avoid the location of strong $XCH_4$ sources as present in the EDGAR inventory (it is a region with an important oil and gas industry). Then areas over Canada were chosen because it represents the strong $XCH_4$ underestimation related to the low surface albedo values in these high latitudes, also present in Northern part of Europe and Russia. Adding areas of Russia did not change the dependence and the fit (Fig. 4) made to derive the correction.

**Comment C 2.15** — On the bias correction method, page 10, line 1. You only apply a bias correction on the surface albedo and say that the other retrieved parameters show negligible dependence, showing surface albedo, AOD and SZA. For OCO-2 the parameter dP (the difference between the a priori and retrieved surface pressure) shows the largest dependence on the bias. Is this a parameter you have looked in to?

**Reply**:

In the TROPOMI $XCH_4$ retrieval we do not retrieve surface pressure. In Fig. 3 in the manuscript we show surface albedo, AOD and SZA as an example, but dependencies in other parameters such as $\chi^2$, column of interfering absorbers $H_2O$ and CO, retrieved aerosol parameters (aerosol size, altitude of aerosol distribution and aerosol column) were also investigated. Besides the fact that we also tried to use as few correction parameters as possible, all the other parameters showed negligible dependence compared to that on surface albedo. We have specified that SZA and AOD are examples on page 10, line 3 to avoid misunderstanding.

**Comment C 2.16** — Page 16. Lines 5-6. Have you tried comparing with snow cover data to verify how reliable this method of detecting snow actually is?

**Reply**: We have not done that comparison ourselves, but we are in contact with colleagues from the Finish Meteorological Institute (FMI) to investigate the seasonality on the bias further. They have found a significant correlation between the seasonality of the bias and the presence of snow surface at Sodankylä, and as the blended albedo is as well correlated to this seasonality, it is suitable to use it to filter this complex scenes, but it is not aimed as an accurate method to actually detect snow. We specify this on page 17, line 1: "By applying it [...] a threshold value of 0.85 is optimal to remove these scenes that cause the seasonality on the bias". However, we do not apply this filter ourselves in an operational mode, as the source of these seasonality of the TROPOMI-TCCON bias is still unknown.

The correlation with snow surface found by FMI does not necessarily always imply the presence of vortex air which was our first hypothesis to explain the seasonality of the bias. We are also investigating the role of the prior profile in this specific retrieval scenarios, assuming that there might be cases with a strong depletion of $XCH_4$ in the upper troposphere (due mainly to vortex air) that impact our retrieval (and/or TCCON) if it is not captured properly by the prior. Furthermore, the different sensitivities between TCCON and TROPOMI might also play a role in the satellite and ground based comparison. All these effects we think need to be taken into account when making conclusions out of the validation results.

**Comment C 2.17** — Page 5, Table 1. Caption missing versions for the instrument.

**Reply**: We assume that this refers to the typo as both instruments are referred as "ll" in the caption of Table 1. The instrument "ll" (Sherlock et al., 2017) was replaced by "lr" (Pollard et al., 2019). We have corrected this.

**Comment C 2.18** — Page 10, line 7. Typo with full stop. I assume you wanted a capital T or a semicolon.

**Reply**: Corrected.

**Comment C 2.19** — There are multiple instances throughout the paper where you've misspelt ppb as pbb. Such as Page 7 line 25, page 9 line 8, page 20 line 16 and twice on page 14 line 4.

**Reply**: We thank the referee for spotting this. We have changed it.

[revised manuscript text omitted]